# Population structure and genetic connectivity of the scalloped hammerhead shark (*Sphyrna lewini*) across nursery grounds from the Eastern Tropical Pacific: Implications for management and conservation

Mariana Elizondo-Sancho[1,2]*, Yehudi Rodríguez-Arriatti[3], Federico J. Albertazzi[4,5], Adrián Bonilla-Salazar[6‡], Daniel Arauz-Naranjo[7‡], Randall Arauz[8,9‡], Elisa Areano[10‡], Cristopher G. Avalos-Castillo[11‡], Óscar Brenes[6‡], Elpis J. Chávez[7‡], Arturo Dominici-Arosemena[3‡], Mario Espinoza[4,9,11], Maike Heidemeyer[5], Rafael Tavares[12‡], Sebastián Hernández[1,13]

1 Laboratorio Biomol, Center for International Programs and Sustainability Studies, Universidad Veritas, San José, Costa Rica, 2 Programa de Posgrado en Biología, Universidad de Costa Rica, San José, Costa Rica, 3 Facultad de Ciencias del Mar, Universidad Marítima Internacional de Panamá (UMIP), La Boca, Corregimiento de Ancón, Panamá, República de Panamá, 4 Escuela de Biología, Universidad de Costa Rica, San José, Costa Rica, 5 Centro de Investigación en Biología Celular y Molecular (CIBCM), Cuidad de la Investigación, Universidad de Costa Rica, Sabanilla Montes de Oca, San José, Costa Rica, 6 Reserva Playa Tortuga, Ojochal, Puntarenas, Costa Rica, 7 CREMA, Centro Rescate de Especies Marinas Amenazadas, San Francisco de Coyote, Guanacaste, Costa Rica, 8 Fins Attached Marine Research and Conservation, Colorado Springs, Colorado, United States of America, 9 MigraMar, Olema, California, United States of America, 10 Fundación Mundo Azul, Departamento de Guatemala, Villa Canales, Guatemala, 11 Centro de Investigación en Ciencias del Mar y Limnología, Universidad de Costa Rica, San José, Costa Rica, 12 Centro para la Investigación de Tiburones (CIT), Distrito Capital, Caracas, Venezuela, 13 Sala de Colecciones, Facultad de Ciencias del Mar, Universidad Católica del Norte, Coquimbo, Chile

☯ These authors contributed equally to this work.
‡ These authors also contributed equally to this work.
* marianaelizancho@gmail.com

## Abstract

Defining demographically independent units and understanding patterns of gene flow between them is essential for managing and conserving exploited populations. The critically endangered scalloped hammerhead shark, *Sphyrna lewini*, is a coastal semi-oceanic species found worldwide in tropical and subtropical waters. Pregnant females give birth in shallow coastal estuarine habitats that serve as nursery grounds for neonates and small juveniles, whereas adults move offshore and become highly migratory. We evaluated the population structure and connectivity of *S. lewini* in coastal areas and one oceanic island (Cocos Island) across the Eastern Tropical Pacific (ETP) using both sequences of the mitochondrial DNA control region (mtCR) and 9 nuclear-encoded microsatellite loci. The mtCR defined two genetically discrete groups: one in the Mexican Pacific and another one in the central-southern Eastern Tropical Pacific (Guatemala, Costa Rica, Panama, and Colombia). Overall, the mtCR data showed low levels of haplotype diversity ranging from 0.000 to 0.608, while nucleotide diversity ranged from 0.000 to 0.0015. More fine-grade population structure was detected using microsatellite loci where Guatemala, Costa Rica, and Panama

**Data Availability Statement:** All the mitochondrial control region sequences are available in Genbank accession numbers: OL692109-OL692337. Microsatellite loci genotypes will be uploaded as a Supporting Information file.

**Funding:** (YR-A) National Secretary of Science and Technology SENACYT (FID-156) https://www.senacyt.gob.pa/en/ (EA/ CA)The Phoenix Zoo (grant project no. 33297) https://www.phoenixzoo.org/ (EA/CA) PADI Foundation (grant no. 32809) http://www.padifoundation.org/ (EA/ CA)Waitt Foundation (grant project no. 33297) https://www.waittfoundation.org/ (EA/CA) Rufford Foundation (grant. no. 22366-1) https://www.rufford.org/ (OB) Fundación Reserva Ojochal https://reservaplayatortuga.org/ (RA) The Whitley Fund for Nature https://whitleyaward.org/ (RA) Sandler Family Foundation https://www.sandlerfoundation.org/ (ME-S) Osa Conservation https://osaconservation.org/ (ME-S) Sistema de Estudios de Posgrado of Universidad de Costa Rica https://www.sep.ucr.ac.cr/ The funders had no role in study design, data collection and analysis, decision to publish, or preparation of the manuscript.

**Competing interests:** The authors have declared that no competing interests exist.

differed significantly. Relatedness analysis revealed that individuals within nursery areas were more closely related than expected by chance, suggesting that *S. lewini* may exhibit reproductive philopatric behaviour within the ETP. Findings of at least two different management units, and evidence of philopatric behaviour call for intensive conservation actions for this highly threatened species in the ETP.

## Introduction

Delimiting demographically independent populations and understanding their levels of genetic diversity and connectivity is central to managing and conserving endangered and exploited species [1–3]. In aquatic ecosystems, animals that occupy high trophic positions generally exhibit high extinction risks due to their large sizes, life-history characteristics, and the exploitation rates they are subjected to [4,5]. Sharks are one of the most threatened groups of marine fishes globally, mainly due to overfishing and habitat degradation which has increased dramatically over the past 20 years [4,6,7]. Population level declines are of major concern in conservation since the effects of genetic drift and inbreeding are pronounced in small populations, which may lead to loss of genetic diversity and compromise the ability of a population to adapt to environmental change [8].

The scalloped hammerhead shark *Sphyrna lewini* (Griffith and Smith, 1834), is a large (up to 420 cm total length, TL), viviparous, coastal semi-oceanic species found worldwide in tropical and sub-tropical waters [9]. Similar to other shark species, *S. lewini*, has low resilience to overfishing due to its slow growth, late sexual maturity, and long gestation periods [10–12]. Throughout its distribution, *S. lewini* has experienced severe population declines [7,13–16], leading to its listing as Critically Endangered by the International Union for the Conservation of Nature (IUCN) Red List of Threatened Species [17]. *S. lewini* has a complex life history in which pregnant females give birth in shallow coastal estuarine habitats that serve as nursery grounds during their early life stages [18,19]. Eventually, large juveniles and adults move offshore and become highly migratory, often schooling around seamounts and near continental shelves [20,21].

The dichotomy between breeding in coastal reproductive habitats and the long-range dispersal of adults displayed by shark species such as *S. lewini* may result in complex population structure [22]. Six distinct population segments of *S. lewini* have been distinguished globally, defined within 1) the North West Atlantic and Gulf of Mexico, 2) Central and South West Atlantic, 3) Eastern Atlantic, 4) Indo-West Pacific, 5) Central Pacific, and 6) in the Eastern Pacific [16,23]. Despite high fishing levels, the Eastern Pacific *S. lewini* distinct population segment has been poorly studied throughout its range. Neonates and juveniles are susceptible to bottom shrimp trawl and small-scale artisanal fisheries inshore [24], whereas adults are a frequent by-catch in pelagic longline and purse-seine fisheries that operate near seamounts and oceanic islands [16,25,26].

Genetically discrete groups are created by reproductive behaviors that segregate populations, which cause allele frequency differentiation through time [27]. Natal philopatry is described as a reproductive behavior in which organisms return to their birthplace to reproduce or give birth [28]. This behavior has been observed in several species of sharks, including the great white shark (*Carcharodon carcharias*), mako shark (*Isurus oxyrinchus*), lemon shark (*Negaprion brevirostris*), blacktip shark (*Carcharhinus limbatus*), sand tiger shark (*Carcharhinus taurus*), speartooth shark (*Glyphis glyphis*), and bull shark (*Carcharhinus leucas*) [29–36].

For a species that uses coastal habitats as nursery areas, such as *S. lewini*, natal philopatry could contribute to the development of genetically discrete groups, where intrinsic reproduction and recruitment may result in population structure at smaller geographic scales than would be expected based on the mobility of the organism [37,38].

To date, studies investigating the genetic structure of *S. lewini* in the Eastern Tropical Pacific ocean (ETP) have been either limited to small geographic areas [7,39] or they have used relatively small sample sizes [40]. Given the limited data on the population structure of *S. lewini* and the high fishing pressure that this species is currently experiencing throughout the ETP, it is important to assess the population structure and genetic diversity in potential nursery areas of the region, to develop effective management and conservation strategies. This study (i) assessed the genetic diversity of *S. lewini* in coastal sites of the ETP, (ii) determined the population structure of *S. lewini* within the ETP, and (iii) evaluated the potential role of natal philopatry in the population dynamics of *S. lewini* within the ETP.

## Materials and methods

### Study region

The study region comprises the majority of the ETP (Fig 1), from the coast of Central America and South America to 140˚W, and from southern Mexico to northern Peru [41]. The ETP includes a complex diversity of coastal environments and oceanic islands with oceanographic conditions that vary seasonally, annually and over longer time scales [42]. Coastal sampling sites were comprised of estuarine systems with predominant mangrove vegetation and muddy coasts 1) Las Lisas and Sipacate in Guatemala (GUA, N = 72); 2) Coyote (COY, N = 34) and Ojochal (OJO, N = 44) in Costa Rica; and 3) Punta Chame in Panama (PAN, N = 65) (Fig 1). Samples were also obtained from an oceanic island in Costa Rica, Cocos Island (ICO, N = 15). Previously collected molecular data from coastal areas in México and Colombia were included in the analysis to cover a broader geographic range. Mexican sites included: Nayarit (NAY), Oaxaca (OAX), Michoacan (MCH), Baja California (BJC), Chiapas (CHP), and Sinaloa [40]; Colombian sites included: Port Buenaventura (PTB), Utria (UTR), Sanquianga (SNQ) and Malpelo Island (MLP) [39] (Fig 1). All samples analyzed were from juveniles except the ones collected in Cocos Island and Malpelo Island which were adults. Sampling sites were plotted using base and raster layers from the Natural Earth (public domain) http://www.naturalearthdata.com/ in ArcMap 10.4 [43] and QGis 2.18.9 [44] (Figs 1, 5 and S2).

### Sample collection

Tissue samples of juvenile *S. lewini* (30–50 cm TL) were collected from artisanal fisheries operating in Costa Rica (N = 78), Panama (N = 65) and Guatemala (N = 72) throughout 2017 and 2018. In addition, samples from adults (63–108 cm TL) were collected opportunistically in Cocos Island (N = 15) with a biopsy dart during scientific cruises conducted in 2008. The use of tissue samples for this study was reviewed by the National Commission for the Management of Biodiversity (CONAGEBIO) of Costa Rica. The technical office of CONAGEBIO emitted the research permit R-CM-VERITAS-001-2021-OT-CONAGEBIO. The Ministry of Environment of Panama issued the research permits SEX/A-61-19 and SEX/A-108-17 and the National Council of Protected Areas of Guatemala issued the research license no. I-DRSO-001-2018. Fin and muscle tissue was preserved in 95% ethanol and stored at -20˚ C. Total DNA was extracted from 25 mg of tissue using the phenol-chloroform protocol [45] and with Promega's Wizard® Genomic DNA Purification Kit.

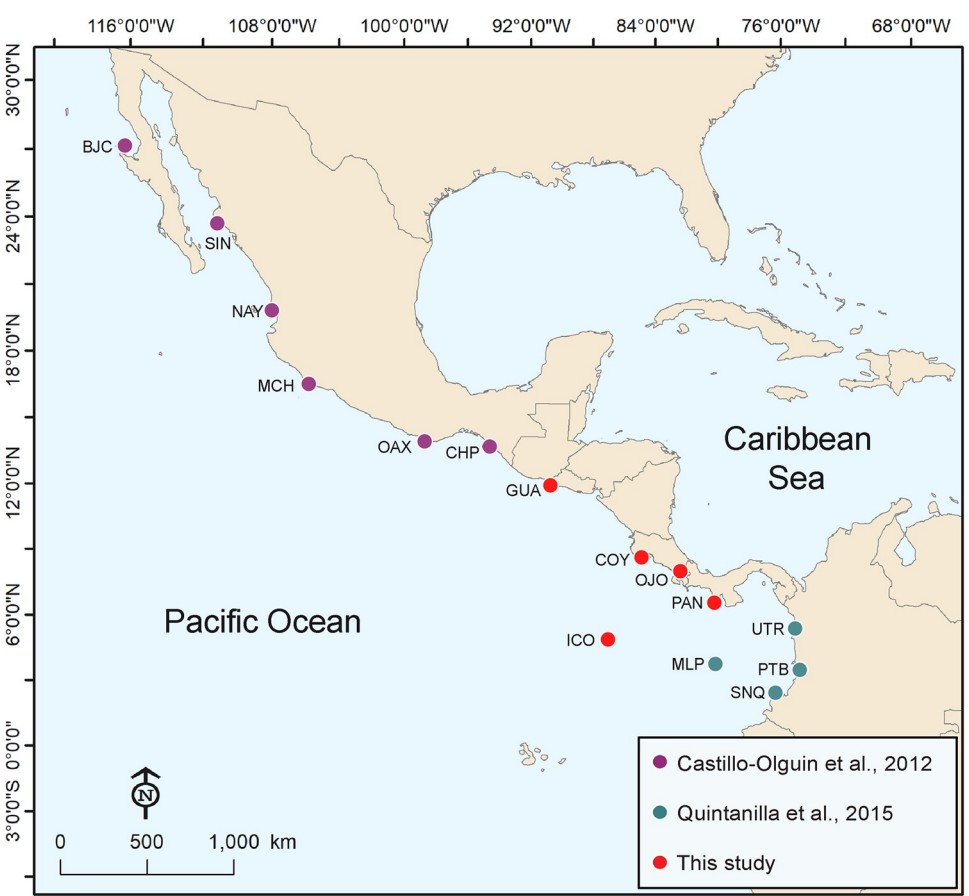

**Fig 1. Location of sampling sites of *Sphyrna lewini* in the Eastern Tropical Pacific.** Sampling sites: Guatemala (GUA, N = 72), Ojochal (OJO, N = 43), Coyote (COY, N = 34), Cocos Island (ICO, N = 15), Panama (PAN, N = 65), Nayarit (NAY, N = 25), Oaxaca (OAX, N = 8), Michoacan (MCH, N = 17), Baja California (BJC, N = 25), Chiapas (CHP, N = 14), Sinaloa (SIN, N = 36), Port Buenaventura (PTB, N = 22), Sanquianga (SNQ, N = 20), Utria (UTR, N = 21), Malpelo Island (MLP, N = 18). Sampling sites were plotted using base and raster layers from the Natural Earth (public domain) http://www.naturalearthdata.com/ in ArcMap 10.4.

### Amplification and sequencing of mitochondrial DNA

The mitochondrial DNA control region (mtCR) was amplified and sequenced for a total of 231 *S. lewini* individuals using primers designed in Geneious Pro v6.0.6 Bioinformatics Software for Sequence Data Analysis [46]. Forward (3′ AAGGGTCAACTTCTGCCCT 5′) and reverse (3′AGCATGGCACTGAAGATGCT 5′) primers were designed based on the whole mitochondrial genome of *S. lewini* deposited in Genbank (Accession number: JX827259). PCR amplification was conducted using a Veriti™ Thermal Block (Applied Biosystems, USA) with a total volume of 15μL containing 67 mM Tris-HCl pH 8.8, 16mM $(NH_4)_2SO_4$, 2.0 mM $MgCl_2$, 20 mM dNTPs, 10 μM of each primer, 0.4 units of Dream Taq DNA Polymerase (5U/ μl), and 1 μl of DNA (20–40 ng/μl). The PCR thermal profile included initial 5 min denaturation at 94˚C, 30 cycles of 30 s at 94˚C, 30 s at 59˚C and 1.5 min at 72˚C, followed by a final extension for 10 min at 72˚C. PCR products and the corresponding negative control were visualized in UV light after electrophoresis in 1.2% agarose gel. PCR products were purified and then sequenced in both directions using an ABI 3100 automated sequencer.

Amplification and genotyping of microsatellite loci.

A total of 169 samples of *S. lewini* were genotyped for 14 microsatellite loci previously described by Nance et al. (2009) (Guatemala = 52; Costa Rica = 50; Panama = 51; Cocos

Island = 15). Forward primers were marked with an M13 tail (5´- TGT AAA ACG ACG GCC AGT-3´) [47]. Microsatellite amplification was conducted using a nested PCR in a total volume of 15 µL with 1–2 µL of DNA (10–30 ng), 0.1 µM forward primer, 0.4 µM reverse primer, 0.4 µM M13 primer (6FAM, VIC or NED), 0.2 mM dNTPs, 2 mM MgCl$_2$, 0.04 units of Dream Taq DNA Polymerase (5U/µL), 1X Buffer and water. PCR conditions consisted of an initial 2 min denaturalization at 94˚C, followed by 32 cycles of 30 s at 94˚C, 30 s of 57˚C (Sle25, Sle77), 59˚C (Sle45, Sle59, Sle33, Sle53), 60˚C (Sle54, Sle13, Sle18, Sle27, Sle81, Sle71, Sle86, Sle38) 1 min a 72˚C, followed by 8 cycles of 30 s at 94˚C, 30 s at 53˚C, 30 s at 72˚C and a final extension of 2 min at 72˚C. PCR products and corresponding negative controls were verified by electrophoresis in 1.2% agarose gel and visualized using UV light. PCR products were cleaned and then sequenced in both directions using an ABI 3730 automated sequencer to verify the microsatellite motifs. Fragment size analysis was done using an ABI 3730 automated sequencer with a 5-dye chemistry and a size standard of GS500.

## Mitochondrial DNA analysis

Collected in this study from coastal sites of the ETP and Cocos Island, 229 sequences from the mtCR of *S. lewini* were analyzed. An alignment of 489bp was carried out with the MUSCLE algorithm on GeneiousPro Bioinformatics Software for Sequence Data Analysis [46]. For a broader geographic range, 206 additional sequences previously published were retrieved from GenBank® (S1 Table) and added to the alignment from the coast of Colombia (N = 81) [39] and México (N = 125) [40]. Arlequin 3.5 Software [48] was used to calculate the number of haplotypes (H), polymorphic sites (S), nucleotide diversity (π), haplotype diversity (hd) and nucleotide base composition. To examine relationships among haplotypes, a haplotype network was drawn in Haploview Bioinformatics software [49] which was based on phylogenetic reconstructions carried out for maximum likelihood in RAxML-HPC2 8.2.12 on XSEDE [50] (available at https://www.phylo.org) [51]. The maximum likelihood analysis was carried out using GTR+ Gamma and 1000 Bootstrap iterations.

Arlequin 3.5 software [48] was used to estimate the population structure among geographic areas using Wright's pairwise fixation index ($\theta_{ST}$) [52] (20,000 permutations, α = 0.05). An exact test of population differentiation based on haplotype frequencies was conducted to complement this analysis using Arlequin 3.5 (100000 steps in the Markov chain, 100000 dememorization steps, α = 0.05) [48]. A global hierarchical Analysis of Molecular Variance (AMOVA) [53] was performed to determine the genetic diversity among and within regions using Arlequin 3.5 (10000 permutations and α = 0.05) [48]. In order to observe which configuration of the data best explained the variance the AMOVA was performed with three different groupings: 1) one region (all locations); 2) two regions the Northern Eastern Tropical Pacific (Mexico) and the Central-southern Eastern Tropical Pacific (Guatemala, Costa Rica, Panama and Colombia); and 3) three regions the Northern Eastern Tropical Pacific (Mexico collection sites), the Central Eastern Pacific (Guatemala, Costa Rica and Panama), and the Southern Eastern Tropical Pacific (Colombia). A Mantel test [54] was conducted to test the hypothesis that genetic differentiation is due to isolation-by-distance; *Adegenet R* package [55] in R v.4.0.2. [56] was used to evaluate the correlation between Nei's genetic distance and a matrix of Euclidian geographic distance.

## Microsatellite DNA data analysis

The fragment size of 14 microsatellite loci for each sample was determined by identifying the peaks with GeneMarker® Software 2.6.3. The presence of genotyping errors and null alleles, as well as the frequency of null alleles per locus (r) was evaluated using MICRO-CHECKER

v.2.2.3 [57]. Deviations from Hardy-Weinberg equilibrium (HW) and linkage disequilibrium (LD) were calculated for each locus and sampling site using GENEPOP v. 4.0 [58] utilizing 10000 steps of dememorization, 1000 batches and 10000 iterations per batch. All probability values were adjusted using the Holm-Bonferroni correction [59]. *Adegenet R* package [55] in R v.4.0.2 [56] was used to calculate the number of alleles (A), allelic richness (Ar), expected heterozygosity (HE), observed heterozygosity (HO) and inbreeding coefficient (FIS).

The package *Related* [60] in R v.4.0.2. [56] was used to conduct the genetic relatedness analysis based on the allele frequencies among all pairs of *S. lewini* individuals within and among sampling sites. The function "comparestimators" was used to select the best relatedness estimator and evaluate the performance of four genetic relatedness estimators [61–64]. This function simulates individuals of known relatedness based on the observed allele frequencies and compares the correlation between observed and expected genetic relatedness for each estimator. The relatedness estimator with the highest correlation coefficient was chosen. To check for the possibility of occurrence of related individuals that may bias estimates of genetic diversity and differentiation, the distribution of observed pairwise relatedness values across all individuals was also compared to the values expected between parent-offspring (PO), full-siblings (FS), half- siblings (HS) and unrelated pairs (U). Subsequently, to determine if individuals within sampling sites were more closely related than expected by chance, observed values of relatedness for each sampling site were compared from random mating expectations with the function "grouprel". This function calculates the average pairwise relationship within each predefined group (i.e., sampling site) as well as an overall within-group relatedness. The expected distribution of average within group relatedness is generated by randomly shuffling individuals using 1000 Monte Carlo simulations, keeping group size constant. The observed mean relatedness is then compared to the distribution of simulated values to test the null hypothesis of groups being randomly associated in terms of relatedness. Additionally, pairwise average relatedness was compared within and between sampling sites with a Two Sample t-test and an alpha threshold of 0.05 in R v.4.0.2. [56].

To test for population structure between sample collection areas, pairwise population comparisons of $D_{EST}$ values [65] and Wright's pairwise fixation index ($F_{ST}$) [52], were obtained using the "fastDivPart" function in the *diveRsity* package [66] in R v.4.0.2. The variance of these statistics was assessed by 10000 bootstrap iterations and a bias corrected 95% confidence interval (CI) was calculated for pairwise calculations [66]. Additionally, the software STRUCTURE v.2.3.4 [67] was used to identify the clustering of groups of individuals and the admixture with a Markov Chain of Monte Carlo (MCMC) (length burn-in period: 200000; MCMC: 40000; 10 K, 10 iterations each). To infer the best K, the Evanno ΔK method was used [68]. To complement previous population structure analysis, a multivariate approach, Discriminant Analysis of Principal Components (DAPC) [69] was used to identify discrete populations based on geographic region, using the *Adegenet* package in R v.4.0.2. The DAPC summarizes initial genetic data into uncorrelated groups using principal components, then uses discriminant analysis to maximize the among-population variation [70]. In the DAPC, retaining too many Principal Component Analysis (PCA) axes with respect to the number of individuals can lead to over-fitting. To decide in an objective way how many PCA axes to retain a cross-validation analysis was performed with the "xvalDapc" function in the *Adegenet* package [69]. This function tries different numbers of PCA axes and then assesses the quality of the corresponding DAPC by cross-validation, with 100 replicas [69]. The number of PCA axes associated with the lowest Mean Squared Error were then retained in the DAPC [69]. Cluster assignments were pre-defined corresponding with defined collection locations.

A Mantel test was performed to test the hypothesis of genetic differentiation due to isolation-by-distance; the correlation between Nei's genetic distance and a matrix of Euclidian

geographic distances were evaluated using the *Adegenet* package [55] in R v.4.0.2. Gene flow was analyzed with the "divMigrate" function [71] of the package *diveRsity* [66] in R v. 4.0.2 [56]. The "divMigrate" function was used to plot the relative migration levels and detect asymmetries in gene flow patterns, between pairs of sampling sites using $D_{EST}$ values of genetic differentiation [71]. This function plots sampling areas connected to every other by two connections that represent the two reciprocal gene flow components between any pair of locations [71]. This approach provides information on the direction of migration using relative migration scales (from 0 to 1) in which the highest migration rate given is one [71].

## Results

### Mitochondrial DNA

The nucleotide alignment (435 sequences and 489pb) of mtCR sequences from individuals across the ETP, had a nucleotide base composition of 31.7% A, 24.4% C, 7.8% G, 36.1% T, 16 haplotypes, and 23 polymorphic sites. Sequences from this study are deposited in Genbank, accession numbers: OL692109—OL692337. There was variation of the genetic diversity of *S. lewini* samples throughout the ETP (Table 1). The haplotype diversity (hd) ranged from 0.000 to 0.608, while nucleotide diversity (π) ranged from 0.000 to 0.0015. The highest genetic diversity was observed in Guatemala (hd = 0.608; π = 0.00015), followed by Malpelo Island (hd = 0.581; π = 0.0012). The lowest genetic diversity was detected in Baja California, Chiapas, and Oaxaca (hd = 0.000, π = 0.000). In all sampling areas the overall haplotype diversity was 0.3912± 0.2215 and the nucleotide diversity 0.0016±0.0016. Overall genetic diversity in the Northern ETP (hd = 0.2175, π = 0.001691) was lower than in the Central-southern ETP (hd = 0.5481, π = 0.001704) (Table 1).

A total of 16 haplotypes were found in all samples across the ETP (Fig 2). Thirteen of these haplotypes were sampled out of two or more individuals where Hap5 was the most common haplotype across all sampling sites and detected in 50.4% of all individuals analyzed (Fig 2 and S3 Table). Two common haplotypes, Hap5 and Hap4 were found across all sampling sites. These two common haplotypes differed in frequency by region, Hap5 was found in higher frequency in the Northern ETP, while Hap4 was found in higher frequency in the Central-Southern ETP. Ten private haplotypes were detected: GUA (4), OJO (2), COY (1), PAN (2), SIN (1).

Pairwise $\theta_{ST}$ values showed significant genetic differentiation between Northern ETP sampling sites (NAY, OAX, MCH, BJC, CHP, SIN) and those in the Central-southern ETP (GUA, OJO, COY, ICO, PAN, PTB, UTR, SNQ and MLP) (Table 2). The hierarchal AMOVA, confirmed this genetic differentiation between samples from the Northern ETP and samples from Central-southern ETP. This configuration of the data was the one that best explained the variation. Significant levels of population subdivision were found between these two groups, representing 37.42% of the variation found in the mtCR (Table 3). The mtCR Mantel test revealed a significant pattern of isolation-by-distance (r = 0.47, p = 0.002), showing that genetic distance was correlated with geographic distance.

### Microsatellite loci

A total of 169 individuals from three coastal sampling sites in Central America (GUA, OJO, and PAN), were genotyped at 14 microsatellite loci. MICRO-CHECKER provided evidence of null alleles on four loci (Sle18, Sle25, Sle53 and Sle77), which were removed from further analysis. Loci Sle13 and Sle27 were found to be linked (p < 0.05, Fisher's method) after performing sampling site-specific and global pairwise comparisons between loci to determine linkage disequilibrium, and the latter was also removed from further analyses. The remaining loci presented no significant deviation from Hardy-Weinberg equilibrium after Holm-Bonferroni

**Table 1. Genetic diversity indices for the mitochondrial control region and 9 microsatellite loci for *Sphyrna lewini* individuals in the Eastern Tropical Pacific.**

| Sites | mtCR | | | | | | 9 Microsatellite loci | | | | | |
|---|---|---|---|---|---|---|---|---|---|---|---|---|
| | n | H | S | hd | Π | Pha | Ho | He | Na | Ua | Fis | Ar |
| **Northern ETP** | 125 | 3 | 14 | 0.2175 | 0.001691 | | | | | | | |
| BJC | 25 | 1 | 0 | 0 | 0 | 0 | | | | | | |
| SIN | 36 | 3 | 14 | 0.300 | 0.005392 | 1 | | | | | | |
| NAY | 25 | 2 | 1 | 0.380 | 0.000757 | 0 | | | | | | |
| MCH | 17 | 2 | 1 | 0.308 | 0.000689 | 0 | | | | | | |
| OAX | 8 | 1 | 0 | 0 | 0 | 0 | | | | | | |
| CHP | 14 | 1 | 0 | 0 | 0 | 0 | | | | | | |
| Overall | 125 | 3 | 14 | 0.2175 | 0.001691 | | | | | | | |
| **Central-southern ETP** | 310 | 15 | 23 | 0.548 | 0.001704 | | | | | | | |
| GUA | 72 | 7 | 4 | 0.608 | 0.001531 | 4 | 0.71 | 0.71 | 8.33 | 10 | 0.03 | 6.74 |
| COY | 34 | 3 | 2 | 0.522 | 0.001141 | 1 | | | | | | |
| OJO | 43 | 6 | 4 | 0.501 | 0.001275 | 2 | 0.68 | 0.72 | 8.67 | 7 | 0.10 | 7 |
| PAN | 65 | 7 | 6 | 0.575 | 0.001433 | 2 | 0.68 | 0.72 | 7.78 | 10 | 0.08 | 5.89 |
| ICO | 15 | 3 | 2 | 0.514 | 0.001246 | 0 | 0.69 | 0.66 | 5.56 | 2 | 0.02 | 6.34 |
| UTR | 21 | 2 | 1 | 0.514 | 0.001052 | 0 | | | | | | |
| PTB | 22 | 4 | 17 | 0.541 | 0.00408 | 0 | | | | | | |
| SNQ | 20 | 3 | 17 | 0.511 | 0.004176 | 0 | | | | | | |
| MLP | 18 | 3 | 2 | 0.581 | 0.001296 | 0 | | | | | | |
| Overall | 125 | 3 | 14 | 0.5481 | 0.001704 | | | | | | | |

N: Number, H: Number of haplotypes, S: Polymorphic sites, hd: Haplotype diversity, π: Nucleotide diversity, Pha: Number of private haplotypes, Ho: Observed heterozygosity, He: Expected heterozygosity, Na: Number of alleles, Ua: Unique alleles, Fis: Inbreeding coefficient, Ar: Allelic richness. Sampling sites: Northern Eastern Tropical Pacific: Baja California (BJC), Sinaloa (SIN), Nayarit (NAY), Michoacan (MCH), Oaxaca (OAX), Chiapas (CHP); and Central-Southern Eastern Tropical Pacific: Guatemala (GUA), Coyote (COY), Ojochal (OJO), Panama (PAN), Cocos Island (ICO), Utria (UTR), Port Buenaventura (PTB), Sanquianga (SNQ), Malpelo Island(MLP).

correction and presented a total of 1.45% of missing data (sample with no interpretable pattern of DNA fragments after PCR amplification). The genetic relatedness estimates of Wang showed the best performance with our data (r = 0.81), demonstrating an overall coefficient of r = -0.06. Individuals sampled across all sites closely followed the distribution of values expected from unrelated pairs (Fig 3). Unique alleles were found in loci Sle013, Sle033, Sle038, Sle054, Sle071, Sle081, Sle086, Sle089 and all sampling sites (Tables 1 and S2).

Genetic diversity metrics were similar between sampling sites (Table 1). The highest values of observed heterozygosity (Ho) and allelic richness (Ar) were found in Guatemala and the lowest in Panama (Table 1). Inbreeding coefficients (Fis) ranged from 0.02 to 0.10 (Table 1). Genetic diversity statistics were similar between the 9 loci analyzed (S2 Table). Allelic richness across loci was 6.13 ± 4.06. The observed heterozygosity (Ho) ranged from 0.539 (Sle054) to 0.835 (Sle089). The inbreeding coefficients (Fis) ranged from 0.003 (Sle045) to 0.074 (Sle033).

Global genetic structure coefficients of $D_{EST}$ and $F_{ST}$ determined significant values of genetic differentiation for *S. lewini* at coastal sampling sites of the ETP ($D_{EST}$ = 0.14, p < 0.05; $F_{ST}$ = 0.054, p < 0.05). Pairwise comparisons of $D_{EST}$ between coastal sampling sites were all significant and showed a greatest differentiation between Guatemala and Panama ($D_{EST}$ = 0.0942, 95% CI (0.0438–0.1123), p < 0.05) and a lowest differentiation for Costa Rica and Panama ($D_{EST}$ = 0.029, 95% CI (0.0096–0.054), p < 0.05). $F_{ST}$ values were concordant with $D_{EST}$ values, showing the least differentiation between Costa Rica and Panama ($F_{ST}$ = 0.0185, 95%

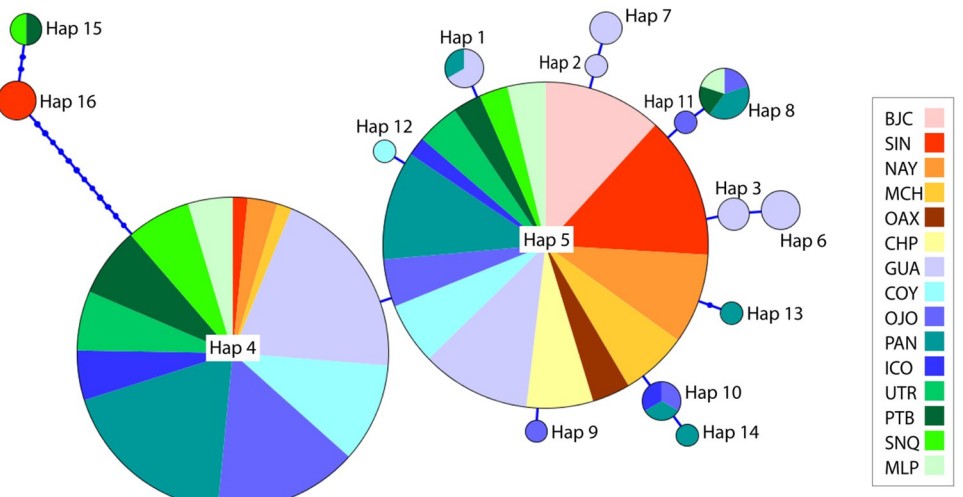

**Fig 2. Haplotype network based on mitochondrial control region sequences for *Sphyrna lewini*.** Each circle represents a unique haplotype (Haplotype 1 through 16); the size of the circle is proportionate to the number of individuals; the colors represent the proportion of individuals from each sampling location; ticks on connecting lines indicate mutational steps between haplotypes. Sampling sites: Guatemala (GUA), Ojochal (OJO), Coyote (COY), Cocos Island (ICO), Panama (PAN), Nayarit (NAY), Oaxaca (OAX), Michoacan (MCH), Baja California (BJC), Chiapas (CHP), Sinaloa (SIN), Port Buenaventura (PTB), Sanquianga (SNQ), Utria (UTR), Malpelo Island (MLP).

CI (0.0089–0.0299), $p < 0.05$) and the most differentiation between Guatemala and Panama ($F_{ST} = 0.0807$, 95% CI (0.0621–0.0985), $p < 0.05$). The DAPC conducted for the coastal sampling sites of the ETP show this same pattern of differentiation (Figs 4A and S1). Two groups were revealed by the STRUCTURE cluster analysis (K = 2), with Costa Rica and Panama

**Table 2. Pairwise $\theta_{ST}$ values and exact test of sample differentiation of the mitochondrial control region for *Sphyrna lewini* individuals in the Eastern Tropical Pacific.**

|  | BJC | SIN | NAY | MCH | OAX | CHP | GUA | COY | OJO | PAN | ICO | UTR | PTB | SNQ | MLP |
|---|---|---|---|---|---|---|---|---|---|---|---|---|---|---|---|
| BJC | - | 0.07072 | 0.02163 | 0.05993 | 0.99995 | 0.99995 | 0.00000 | 0.00000 | 0.00000 | 0.00000 | 0.00000 | 0.00000 | 0.00000 | 0.00000 | 0.00000 |
| SIN | 0.049 | - | 0.17235 | 0.54013 | 0.43470 | 0.18609 | 0.00000 | 0.00000 | 0.00000 | 0.00000 | 0.00000 | 0.00015 | 0.00000 | 0.00000 | 0.00000 |
| NAY | **0.208*** | 0.027 | - | 0.71997 | 0.29791 | 0.07127 | 0.00045 | 0.00729 | 0.00020 | 0.00150 | 0.00400 | 0.03456 | 0.00265 | 0.00405 | 0.05743 |
| MCH | 0.169 | 0.009 | -0.039 | - | 0.52250 | 0.22344 | 0.00065 | 0.00459 | 0.00005 | 0.00190 | 0.00355 | 0.02003 | 0.00135 | 0.00280 | 0.03586 |
| OAX | 0.000 | -0.015 | 0.101 | 0.050 | - | 0.99995 | 0.00040 | 0.00285 | 0.00005 | 0.00045 | 0.00105 | 0.00924 | 0.00095 | 0.00215 | 0.00215 |
| CHP | 0.000 | 0.020 | 0.151 | 0.105 | 0.000 | - | 0.00000 | 0.00015 | 0.00000 | 0.00000 | 0.00010 | 0.00060 | 0.00000 | 0.00005 | 0.00105 |
| GUA | **0.395*** | **0.175*** | **0.163*** | **0.205*** | **0.329*** | **0.359*** | - | 0.64056 | 0.18823 | 0.71977 | 0.55986 | 0.63117 | 0.56540 | 0.59197 | 0.53264 |
| COY | **0.510*** | **0.134*** | **0.179*** | **0.239*** | **0.403*** | **0.449*** | -0.009 | - | 0.24562 | 0.86960 | 0.52060 | 0.99995 | 0.51850 | 0.68551 | 0.66888 |
| OJO | **0.549*** | **0.194*** | **0.270*** | **0.323*** | **0.461*** | **0.499*** | 0.007 | 0.001 | - | 0.21111 | 0.99995 | 0.24382 | 0.93747 | 0.79918 | 0.16386 |
| PAN | **0.406*** | **0.163*** | **0.158*** | **0.203*** | **0.335*** | **0.367*** | -0.002 | -0.017 | -0.003 | - | 0.58757 | 0.85432 | 0.61814 | 0.63312 | 0.71578 |
| ICO | **0.661*** | **0.123*** | **0.258*** | **0.325*** | **0.502*** | **0.575*** | -0.020 | -0.029 | -0.037 | -0.031 | - | 0.48160 | 0.99995 | 0.99995 | 0.36678 |
| UTR | **0.575*** | **0.103*** | **0.170*** | **0.240*** | **0.435*** | **0.497*** | -0.018 | -0.038 | -0.003 | -0.027 | -0.034 | - | 0.52215 | 0.51755 | 0.86401 |
| PTB | **0.366*** | **0.106*** | **0.169*** | **0.181*** | **0.231*** | **0.291*** | 0.026 | 0.014 | -0.001 | 0.019 | -0.025 | 0.003 | - | 0.99995 | 0.35974 |
| SNQ | **0.352*** | **0.090*** | **0.153*** | **0.165*** | **0.211*** | **0.273*** | 0.022 | 0.009 | 0.002 | 0.016 | -0.027 | -0.004 | -0.049 | - | 0.40434 |
| MLP | **0.532*** | **0.086*** | 0.141 | 0.201 | **0.377*** | **0.446*** | -0.016 | -0.035 | -0.005 | -0.030 | -0.030 | -0.048 | -0.005 | -0.008 | - |

Sampling sites: Sampling sites: Baja California (BJC), Sinaloa (SIN), Nayarit (NAY), Michoacan (MCH), Oaxaca (OAX), Chiapas (CHP), Guatemala (GUA), Coyote (COY), Ojochal (OJO), Panama (PAN), Cocos Island (ICO), Utria (UTR), Port Buenaventura (PTB), Sanquianga (SNQ), Malpelo Island (MLP). Significant values ($p < 0.05$) are found in bold letters, significant values ($p < 0.05$) of exact tests of sample differentiation based on haplotype frequencies are represented with an asterisk (*). P values of the Pairwise $\theta_{ST}$ analysis are presented above the diagonal.

**Table 3. Hierarchical Analysis of Molecular Variance (AMOVA) on sequences of the mitochondrial control region for *Sphyrna lewini* in the Eastern Tropical Pacific.**

| | DF | SSD | VC | %V | θ Statistic | P value |
|---|---|---|---|---|---|---|
| **One region** (BJC,SIN,NAY,MCH,OAX,CHP,GUA,COY, OJO,PAN, ICO, UTR,PTB,SNQ,MLP) | | | | | | |
| Among populations | 14 | 26.850 | 0.05990 | **20.83** | | |
| Within populations | 420 | 95.640 | 0.22771 | 79.17 | $\theta_{ST} = 0.2082$ | 0.00000+-0.00000 |
| **Two regions** (BJC,SIN,NAY,MCH,OAX,CHP) (GUA,COY,OJO,PAN,ICO,UTR,PTB,SNQ,MLP) | | | | | | |
| Among regions | 1 | 24.321 | 0.13544 | **37.42** | $\theta_{CT} = 0.374$ | 0.00005+-0.00005 |
| Among populations within regions | 13 | 2.529 | -0.00119 | -0.33 | $\theta_{SC} = -0.0052$ | 0.54061+-0.00344 |
| Within populations | 420 | 95.640 | 0.22771 | 62.91 | $\theta_{ST} = 0.37417$ | 0.00000+-0.00000 |
| **Three regions** (BJC,SIN,NAY,MCH,OAX,CHP) (GUA,COY,OJO,PAN,ICO) (UTR,PTB,SNQ,MLP) | | | | | | |
| Among regions | 2 | 24.414 | 0.09115 | **28.67** | $\theta_{CT} = 0.2866$ | 0.00000+-0.00000 |
| Among populations within regions | 12 | 2.435 | -0.00089 | -0.28 | $\theta_{SC} = -0.0039$ | 0.44446+-0.00362 |
| Within populations | 420 | 95.640 | 0.22771 | 71.61 | $\theta_{ST} = 0.2838$ | 0.00000+-0.00000 |
| **Total** | **434** | **122.490** | | | | |

DF: Degrees of freedom, SSD sum of squares, VC variance component, and % V percent of variance. A comparison of different genetic groupings is presented in the following way: 1) one region: (all locations); 2) two regions: Northern Eastern Tropical Pacific (Mexico) and Central-southern Eastern Tropical Pacific (Guatemala, Costa Rica, Panama and Colombia); and 3) three regions: Northern Eastern Tropical Pacific (Mexico collection sites), Central Eastern Pacific (Guatemala, Costa Rica and Panama), and Southern Eastern Tropical Pacific (Colombia).

conforming one genetic cluster and Guatemala another (Fig 4B). STRUCTURE graphical visualizations with different values of K, reveal the pattern of three distinct groups with Costa Rica and Panama having a higher level of admixture (Fig 4B). Genetic distance was not correlated to the geographic distance between sites since the Mantel test revealed no significant IBD (p > 0.05.) Analysis of the extent and direction of gene flow showed no significant asymmetric movement between coastal sampling sites. However, relative pairwise gene flow demonstrated higher connectivity between Costa Rica and Panama than between Panama and Guatemala (S2 Fig). The genetic exchange obtained with this analysis coincides with the genetic population structure found in pairwise fixation indexes ($D_{EST}$ and $F_{ST}$) and the cluster analyses (STRUCTURE and DAPC) (Fig 4). Gene flow analysis, pairwise fixation indexes ($D_{EST}$ and $F_{ST}$) and cluster analyses (STRUCTURE and DAPC) including Cocos Island, demonstrated higher connectivity of this oceanic island with Costa Rica and Panama than with Guatemala (Figs 4C, 4D and 5 and S4 Table). Pairwise fixation indexes ($D_{EST}$ and $F_{ST}$), show Cocos Island is not significantly differentiated from Costa Rica and Panama.

Overall observed average relatedness calculated from Wang (R = 0.0079) was significantly higher than would be expected by chance (R = -0.0391), indicating non-random relatedness in *S. lewini* individuals within sampling site (S3 Fig). Additionally, the observed average relatedness in each sampling site was significantly higher than expected, indicating that individuals from these areas were more closely related within sampling sites than would be expected by chance (S3 Fig). The distribution of pairwise relatedness calculated from Wang, tends to be higher between individuals within sampling sites than between individuals in different sampling areas (Fig 6). Within each sampling area, the mean pairwise relatedness differed significantly from that found between sampling areas (Two sample t test, t = 24.326, df = 11626, P = 2.2e-16) (S4 Fig). No difference in average pairwise relatedness was observed between males and females (S5 Fig).

## Discussion

Unravelling the genetic structure of threatened or exploited marine species is a critical step in developing more effective management and conservation approaches. This is the first study to

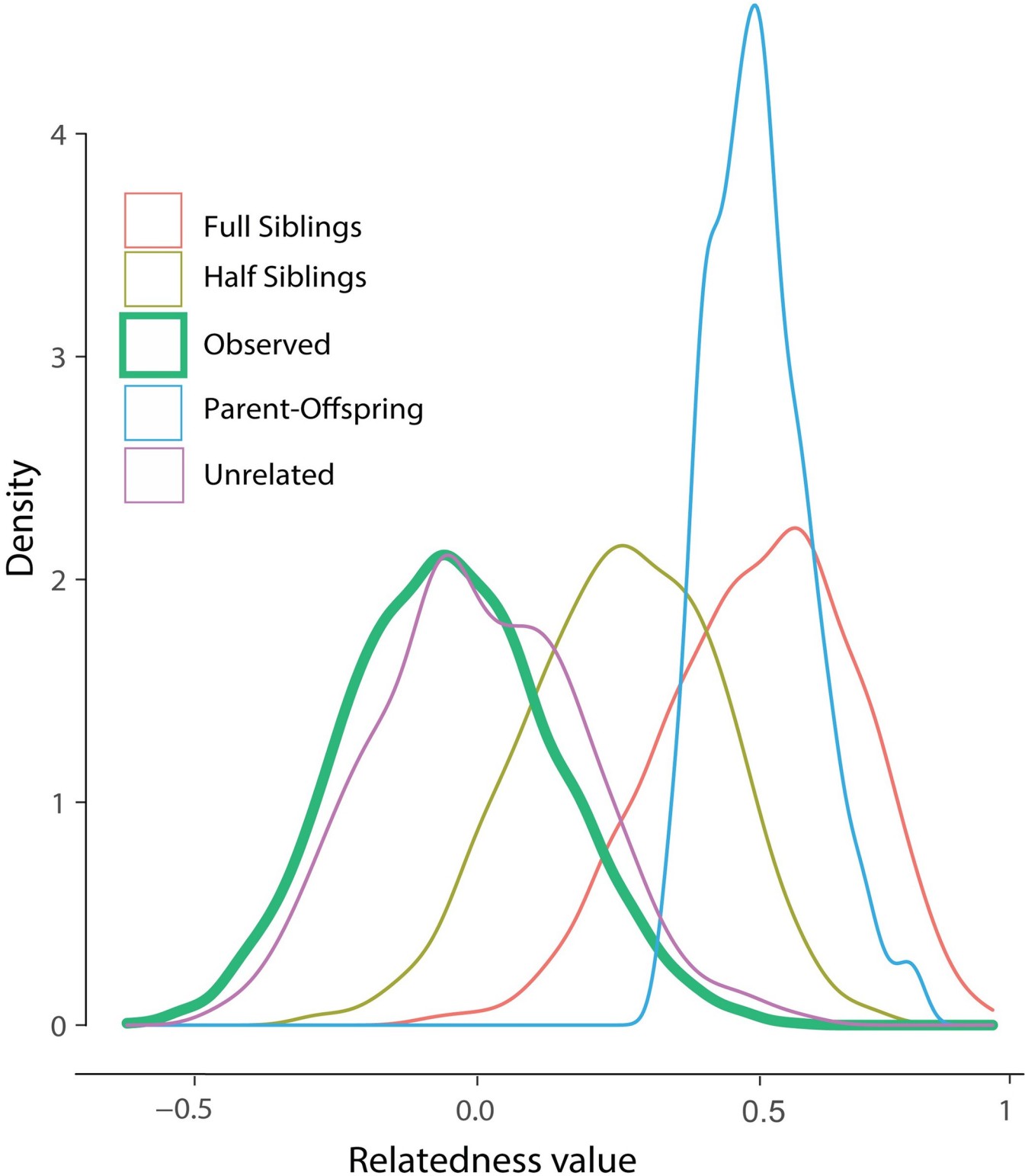

**Fig 3. Distribution of pairwise genetic relatedness.** Distribution of pairwise genetic relatedness values for simulated pairs of individuals: Full siblings (FS), Half siblings (HS), Parent/Offspring (PO), and for observed pairs of individuals of *Sphyrna lewini* sampled in coastal areas of the ETP.

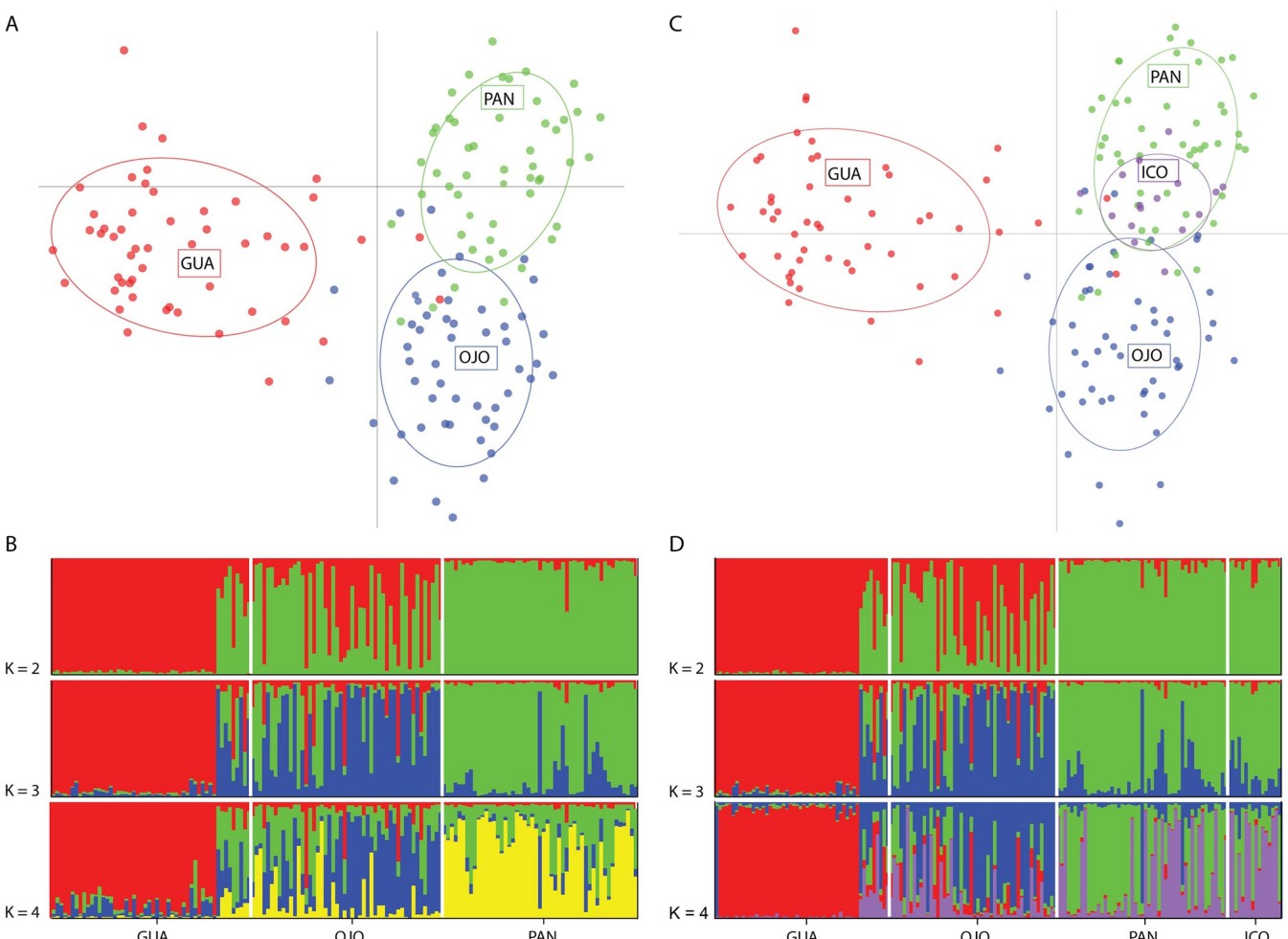

**Fig 4. Population structure analyses from microsatellite genotypes of *Sphryna lewini* individuals in sampling sites of the Eastern Tropical Pacific: Guatemala (GUA), Costa Rica (OJO), Panama (PAN), and Cocos Island (ICO). A)** DAPC plot from the first and second components of the nuclear microsatellite genotypes of three coastal areas **B)** Genetic clusters inferred by STRUCTURE with K = 2, K = 3 and K = 4 of three coastal areas. **C)** DAPC plot from the first and second components of the nuclear microsatellite genotypes of three coastal areas and an oceanic island. **D)** Genetic clusters inferred by STRUCTURE with K = 2, K = 3 and K = 4 of three coastal areas and an oceanic island.

use a robust sample size to examine the fine-scale population genetic structure of this critically endangered shark species throughout the ETP. Patterns of genetic variation of *S. lewini* across coastal areas and an oceanic island within the ETP were assessed using both nuclear-encoded microsatellites and sequences of the maternally inherited mtCR.

## Genetic diversity

As with previous analyses of mtCR in the ETP [7,39,40,72], low levels of genetic diversity for *S. lewini* were found (hd = 0.391). These levels of mitochondrial genetic diversity are comparable to those found in a recent study of this species in the Central Pacific Ocean (hd = 0.439) and are lower than in the Central Indo-Pacific (hd = 0.835) and the western Indian Ocean (hd = 0.653) [72]. The low levels of diversity found in the ETP compared to other geographic locations, are consistent with the evidence suggesting *S. lewini* center of origin was likely from the Indo-Pacific Ocean [9]. Regions as the ETP could have been colonized taking a small

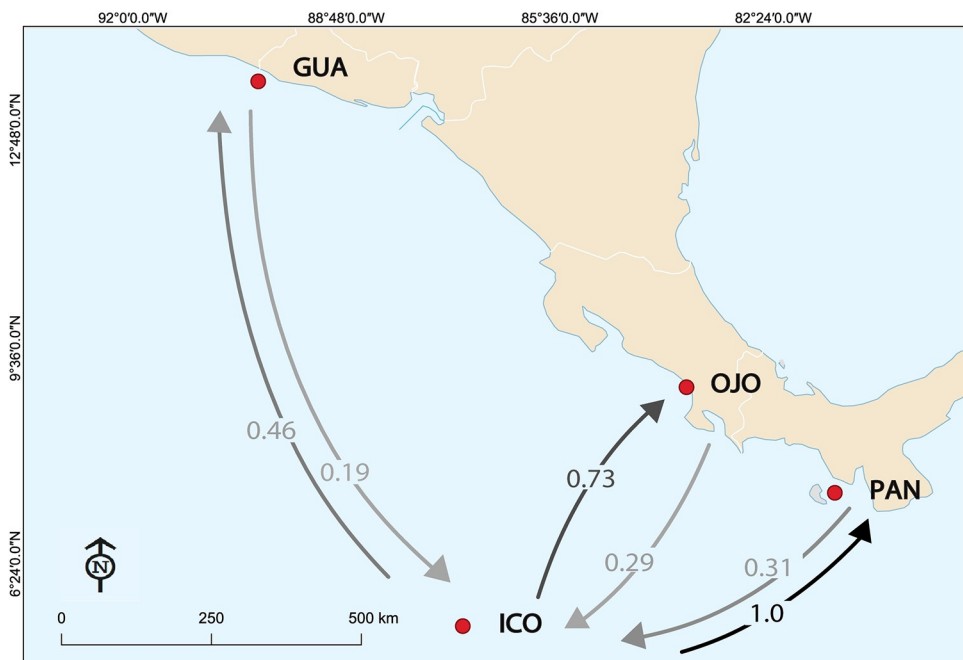

**Fig 5. Contemporary gene flow estimated from 9 microsatellite loci genotypes with the divMigrate function.**
Arrows represent the relative number of migrants and estimated direction of gene flow between three coastal areas:
Guatemala (GUA), Costa Rica (OJO), Panama (PAN); and an oceanic island: Cocos Island (ICO). The darker the
arrow, the higher the relative number of migrants between sampling locations.

sample of the diversity from the source population and consequently experienced strong
genetic drift that promoted the fixation of haplotypes. Comparing these levels of genetic diver-
sity with other sphyrnids, the bonnethead shark (*Sphyrna tiburo*) showed much higher levels
in the Atlantic Ocean (hd = 0.92) [73], as well as the smooth hammerhead shark (*Sphyrna
zygaena*) in the Northern Mexican Pacific Ocean (hd = 0.86) [74] and in the Southern Pacific
Ocean (hd = 0.55) [75]. In this study, gene diversity based on nucleotide and haplotype diver-
sity, was highest in the Central-southern ETP with 15 haplotypes resolved, and lowest in the
Northern ETP with only three haplotypes present. Low genetic diversity has been previously
attributed to overexploitation of this species [74], nonetheless sharks have some of the slowest
mutational rates among vertebrates, so genetic diversity accumulates slowly even in the
absence of population declines [76,77]. Given the long generation time of *S.lewini*, and the rel-
atively short time this species has been prone to overexploitation, most likely the genetic diver-
sity has been shaped by other historical demographic events. Future studies could analyze with
high resolution, greater portions of the genome to see if this low genetic diversity observed
indeed reflects other historical processes.

In addition, nuclear microsatellite marker's observed heterozygosity showed similar values
to those previously reported for this species in the region Ho = 0.703 [22], Ho = 0.770 [7].
Observed heterozygosity in the ETP is similar to that reported for *S. lewini* in the Indian
Ocean (Ho = 0.729) [22] and is higher than the heterozygosity values found in the Western
Atlantic Ocean [Ho = 0.580) [70]. When comparing the nuclear genetic diversity found in this
study with that of other species, the values are similar to those reported for coastal sharks,
including the bonnethead shark (*S. tiburo*) (Ho = 0.59–0.69; [78]) and the blacknose shark
(*Carcharhinus acronotus)* (Ho = 0.66–68; [79]).

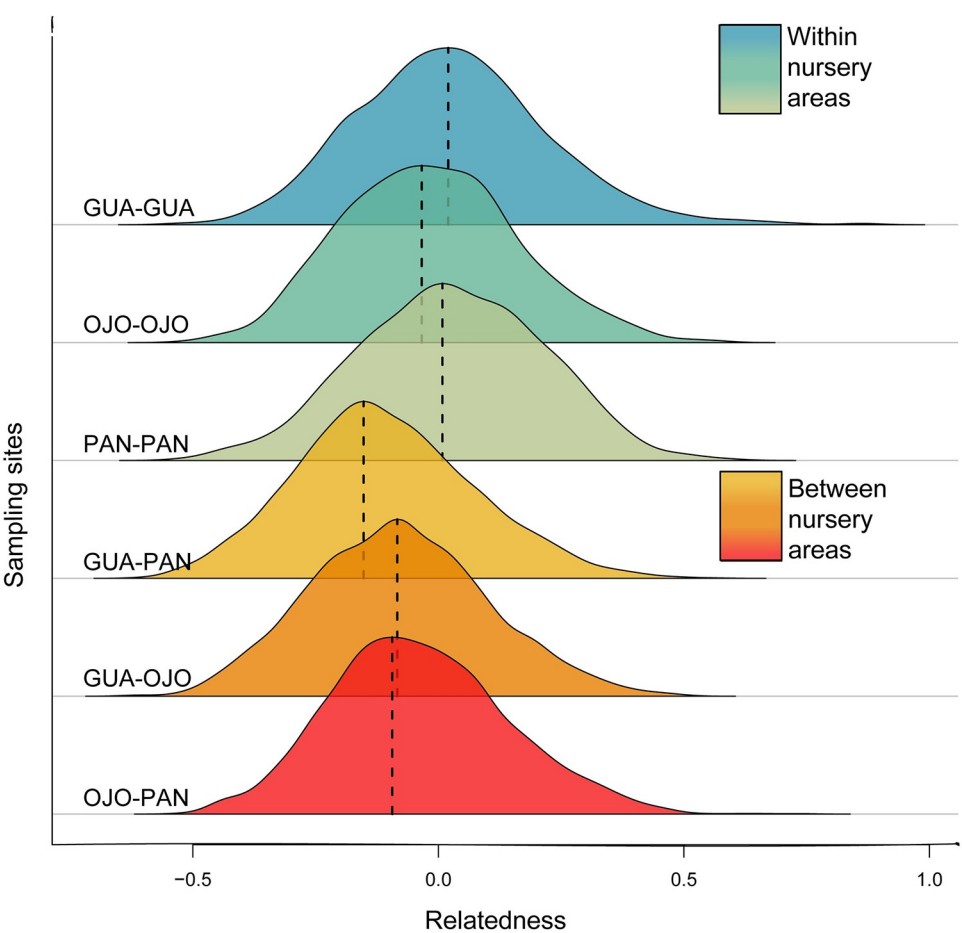

**Fig 6. Distribution of pairwise relatedness values of the Wang estimator of *Sphyrna lewini* individuals within same sampling sites and between different sampling sites.** The mode of each distribution is presented in a black dashed line.

## Population genetic structure

The mitochondrial DNA haplotype distribution of *S. lewini* revealed a pattern of differentiation between the Northern ETP and the Central-southern ETP. This pattern is mainly due to an uneven distribution of the two most common haplotypes, one is found in higher frequency in the Northern ETP while the other is found in higher frequency in the Central-southern ETP. These results differ from the genetic homogeneity that has been previously observed for *S. lewini* in the ETP [7,9], which may be partially explained by the finer geographic sampling and larger sampling sizes used in this study. The entire Eastern Pacific is considered as a single, well-defined distinct population segment of *S. lewini* [16,23], yet based on our findings, this definition should be re-evaluated. Additionally, the low levels of mtDNA polymorphism observed suggests that the mtCR variation in *S. lewini* is insufficient to detect genetic heterogeneity at small scales. It is possible that using more mitochondrial regions or the complete mitogenome could provide a higher resolution, as demonstrated in the speartooth shark (*Glyphis glyphis*), the bull shark (*Carcharhinus leucas*) and the silky shark (*Carcharhinus falciformis*) [35,76,80].

The genetic break identified in our study is located between the boundaries of the Costa Rica Dome and the Tehuantepec Bowl [81], suggesting that the seasonal dynamics of these

systems generate oceanographic conditions that may have an impact on gene flow for *S. lewini* and other marine species [82]. In the ETP, Rodriguez-Zarate et al. (2018) detected a similar pattern of genetic differentiation in the mtCR of the olive ridley sea turtle (*Lepidochelys olivacea*), a migratory marine species with similar life history traits as *S. lewini*, where Mexican nesting colonies were genetically differentiated from those in Central America. Their study determined the existence of two oceanographically dynamic but disconnected regions in the ETP, with a physical mixing zone located in southern Mexico [82]. Pazmiño et al. (2018), also detected differentiation within the ETP region separating the galapagos shark (*Carcharhinus galapagensis*) mtCR sequences found in the Galapagos Islands from the mtCR sequences found in Mexico; this pattern was attributed to secondary barriers that have generated historical geographic isolation [83]. A recent study on *S. tiburo*, a species that is closely related to *S. lewini*, shows that magnetic map cues can elicit homeward orientation [84]. This map-like use of the information of Earth's magnetic field offers a new explanation on how migratory routes and population structure of sharks can be maintained in marine environments.

The genetic differentiation tests ($D_{EST}$ and $F_{ST}$) based on nuclear microsatellite loci revealed three genetically independent units: Guatemala, Costa Rica, and Panama, with limited gene flow between these coastal areas. Despite the limited gene flow found between the three coastal areas, the greatest genetic similarity is observed between Costa Rica and Panama's demes. Graphical representations of clustering analyses (DAPC and STRUCTURE) in Guatemala, Costa Rica and Panama, revealed three distinct groups yet Costa Rica and Panama appear closer together and present more admixture. Additionally, STRUCTURE analyses present K = 2 as the best clustering assignment of the data, with Costa Rica and Panama representing one genetic group, distinct from Guatemala. Connectivity was detected between Cocos Island and the three coastal areas, and more gene flow is observed between this oceanic island and Costa Rica and Panama than with Guatemala. This is the first observation of genetic connectivity between Cocos Island and coastal areas of Central America, and is analogous to the gene flow found between the oceanic island of Malpelo and coastal areas of the Colombian Pacific [39]. The observations of genetic differentiation between coastal nursery areas together with the genetic connectivity with oceanic aggregation areas of adults, suggest that *S. lewini* exhibits philopatry to specific coastal areas in the ETP region. Adult females may undertake long-range migrations to oceanic islands within the ETP but return to specific parturition areas.

## Relatedness and natal philopatry

Inferring relatedness from genotypic data of few loci, remains a challenge and should be used with caution [85,86], nevertheless it provides insight into the potential mechanisms underlying fine-scale behavioral processes with long term consequences on population dynamics. Female fidelity to specific nurseries may define reproductive units if females are returning to the same location during each gestation cycle to give birth, leading to closer relatedness among juveniles from the same location than with individuals from surrounding areas [35]. Individuals within nursery areas were found to be more closely related than expected by chance, thus suggesting that *S. lewini* may exhibit reproductive philopatric behavior within the ETP. This behavior could explain the significant difference in the mean relatedness observed within nursery areas when compared to that found between nursery areas.

Given that *S. lewini* can undertake long-range migrations within the ETP [87], it can be inferred that the resulting population structure is not a consequence of limited dispersal ability. Moreover, all our sampling sites are potential nurseries for *S. lewini* in the ETP and the observed nuclear genetic structure does not support the relation of increased genetic differentiation with increasing geographic distance. This pattern has also been observed in the Atlantic

Ocean, where the main factor driving population subdivision in *S. lewini* is reproductive philopatric behavior rather than oceanographic or geophysical barriers [70].

## Implications for conservation and management

These results offer new insights into the genetic diversity and connectivity of *S. lewini* in the ETP. Our fine-scale population genetic analysis revealed the existence of at least two genetically distinct units within the ETP, one in the Northern ETP and another one in the Central-southern ETP. The strong genetic partitioning found, urges the recognition of two different management units in the ETP; a region that was previously considered to be one distinct population segment of *S. lewini* [16,23]. The low levels of genetic diversity found in *S. lewini* individuals of the Northern ETP, call for special attention to this region. Additionally, coastal sites from Guatemala, Costa Rica and Panama were found to have different evolutionary dynamics, probably attributable to female philopatry. Recent studies of *S. lewini* using genomic data have found finer scale structure than previously documented using genetic data [72]. The question of there being further structuring in the ETP region, should be addressed with higher resolution genetic techniques that could correctly identify discrete population subdivision.

The potential presence of philopatric behavior of *S. lewini* within the ETP emphasizes the need to develop more effective conservation approaches. All coastal sites along the ETP that could potentially serve as nursery areas for *S. lewini* are currently subject to illegal, unreported and unregulated fishing [39,88,89]. Therefore, protection of these nursery areas is crucial for maintaining the genetic diversity, and consequently adaptive potential, of this critically endangered species [1]. For a philopatric species, management measures that identify and protect parturition areas, migratory routes, and unique localized genetic diversity could be crucial to avoid local extinctions [37].

## Supporting information

**S1 Fig. Densities of individuals in discriminant function 1 of 9 nuclear microsatellite loci genotypes of *Sphyrna lewini* in three collection areas of Eastern Tropical Pacific: Guatemala (GUA), Costa Rica (OJO), Panama (PAN).**
(TIF)

**S2 Fig. Contemporary gene flow estimated from 9 microsatellite loci genotypes with the divMigrate function.** Arrows represent the relative number of migrants and estimated direction of gene flow between Guatemala (GUA), Costa Rica (OJO), and Panama (PAN). The darker the arrow, the higher the relative number of migrants between sampling locations.
(TIF)

**S3 Fig. Observed and expected distribution of average relatedness.** Expected distribution of average relatedness based on the Wang estimator of *Sphyrna lewini* in each sampling site and overall sampling sites using 1000 iterations. The average relatedness observed within sampling site and overall sampling site is the statistic test (observed in a red arrow). The further away the statistic test is from the simulated bars, the greater the significance of the relatedness test.
(TIF)

**S4 Fig. Distribution of the pairwise relatedness values of the Wang estimator of *Sphyrna lewini* within sampling sites and between sampling sites.**
(TIF)

**S5 Fig. Distribution of the pairwise relatedness values of the Wang estimator in females and males of _Sphyrna lewini_ overall sampling sites.**
(TIF)

**S1 Table. Localities, the total number (n) and accession number of mitochondrial control region gene sequences for _Sphyrna lewini_ from the Eastern Tropical Pacific.**
(PDF)

**S2 Table. Genetic diversity indices of each microsatellite loci from _Sphyrna lewini_ individuals in the Eastern Tropical Pacific.** Ta: Annealing temperature, Ho: Observed heterozygosity, He: Expected heterozygosity, Ar: Allelic richness, Na: Number of alleles, Ua: Unique alleles, Fis: Inbreeding coefficient.
(PDF)

**S3 Table. Geographic distribution and frequency of mitochondrial control region haplotypes of _Sphyrna lewini_ individuals from the Eastern Tropical Pacific.**
(PDF)

**S4 Table. Pairwise fixation indices ($D_{EST}$ and $F_{ST}$) with lower and upper 95% confidence intervals (CI), between sampling areas of the Eastern Tropical Pacific.** Significant values $\alpha = 0.05$, are presented in bold.
(PDF)

**S1 File.**
(XLSX)

## Acknowledgments

We thank the National Council of Protected Areas of Guatemala for issuing the research license (license no. I-DRSO-001-2018), the National Secretary of Science and Technology of Panama (SENACYT), the Ministry of Environment of Panama for the research permits (SEX/A-61-19 and SEX/A-108-17), the fishers from the community of Las Lisas Guatemala, Daniel Góngora and other fishers from the Punta Chame community for help in field work, Regina Domingo for sample collection in Punta Chame market, Alejandra Barahona director of the Center for International Programs and Sustainability Studies from Universidad Veritas for her support, members of Shark Defenders, small scale fishers of Coyote and Bejuco, in Nandayure, Guanacaste, Costa Rica, The Alvaro Ugalde Scholarship issued by Osa Conservation, and CONAGEBIO in Costa Rica for the research permits (CM-VERITAS-001-2021).

## Author Contributions

**Conceptualization:** Mariana Elizondo-Sancho, Federico J. Albertazzi, Mario Espinoza, Maike Heidemeyer, Sebastián Hernández.

**Data curation:** Mariana Elizondo-Sancho, Maike Heidemeyer, Sebastián Hernández.

**Formal analysis:** Mariana Elizondo-Sancho.

**Funding acquisition:** Mariana Elizondo-Sancho, Yehudi Rodríguez-Arriatti, Federico J. Albertazzi, Randall Arauz, Elisa Areano, Maike Heidemeyer, Sebastián Hernández.

**Investigation:** Mariana Elizondo-Sancho, Yehudi Rodríguez-Arriatti, Adrián Bonilla-Salazar, Daniel Arauz-Naranjo, Randall Arauz, Elisa Areano, Cristopher G. Avalos-Castillo, Óscar Brenes, Elpis J. Chávez, Sebastián Hernández.

**Methodology:** Mariana Elizondo-Sancho, Federico J. Albertazzi, Mario Espinoza, Sebastián Hernández.

**Project administration:** Mariana Elizondo-Sancho, Yehudi Rodríguez-Arriatti, Randall Arauz, Elisa Areano, Arturo Dominici-Arosemena, Maike Heidemeyer, Rafael Tavares, Sebastián Hernández.

**Resources:** Mariana Elizondo-Sancho, Yehudi Rodríguez-Arriatti, Federico J. Albertazzi, Maike Heidemeyer, Sebastián Hernández.

**Supervision:** Federico J. Albertazzi, Mario Espinoza, Sebastián Hernández.

**Visualization:** Mariana Elizondo-Sancho, Mario Espinoza.

**Writing – original draft:** Mariana Elizondo-Sancho, Mario Espinoza.

**Writing – review & editing:** Mariana Elizondo-Sancho, Yehudi Rodríguez-Arriatti, Federico J. Albertazzi, Adrián Bonilla-Salazar, Daniel Arauz-Naranjo, Randall Arauz, Elisa Areano, Cristopher G. Avalos-Castillo, Óscar Brenes, Elpis J. Chávez, Arturo Dominici-Arosemena, Mario Espinoza, Maike Heidemeyer, Rafael Tavares, Sebastián Hernández.

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
