## [Decision Letter · Decision Letter 0]

19 Apr 2022

PONE-D-22-04909Population structure and genetic connectivity of the scalloped hammerhead shark (Sphyrna lewini) across nursery grounds from the Eastern Tropical Pacific: implications for management and conservationPLOS ONE

Dear Dr. Elizondo-Sancho,

Thank you for submitting your manuscript to PLOS ONE. After careful consideration, we feel that it has merit but does not fully meet PLOS ONE’s publication criteria as it currently stands. Therefore, we invite you to submit a revised version of the manuscript that addresses the points raised during the review process.

Thank you for submitting your manuscript PONE-D-22-04909 “Population structure and genetic connectivity of the scalloped hammerhead shark (Sphyrna lewini) across nursery grounds from the Eastern Tropical Pacific: implications for management and conservation” to PLOS ONE. 

I have now received feedback from three experts in the field. As you can see below, they all found the paper of interest but had a few comments. In particular they found that some details on the analyses are missing which could make the paper more robust and understandable. They provided constructive comments that will help you to improve your paper. 

As a result, I invite you to resubmit a revised version of the paper addressing the comments made by these referees.

With kind regards,

Johann 

We look forward to receiving your revised manuscript.

Kind regards,

Johann Mourier, Ph.D.

Academic Editor

PLOS ONE

Journal Requirements:

"This project was funded by: The National Secretary of Science and Technology SENACYT (FID-156) executed by the Ramsar Regional Center for Training and Research on Wetlands in the Western (CREHO), the PADI Foundation (grant no. 32809), The Phoenix Zoo (grant project no. 33297), the Waitt Foundation (grant project no. 33297), the Rufford Foundation (grant. No. 22366-1), Fundación Reserva Ojochal, The Whitley Fund for Nature, Sandler Family Foundation, Osa Conservation, and Sistema de Estudios de Posgrado of Universidad de Costa Rica."

We note that you have provided funding information. However, funding information should not appear in the Funding section or other areas of your manuscript. We will only publish funding information present in the Funding Statement section of the online submission form. 

"(YR-A) National Secretary of Science and Technology SENACYT (FID-156) https://www.senacyt.gob.pa/en/

(EA/ CA)The Phoenix Zoo (grant project no. 33297) https://www.phoenixzoo.org/

(EA/CA) PADI Foundation (grant no. 32809) http://www.padifoundation.org/

(EA/ CA)Waitt Foundation (grant project no. 33297) https://www.waittfoundation.org/

(EA/CA) Rufford Foundation (grant. no. 22366-1) https://www.rufford.org/

(OB) Fundación Reserva Ojochal https://reservaplayatortuga.org/

(RA) The Whitley Fund for Nature https://whitleyaward.org/

(RA) Sandler Family Foundation https://www.sandlerfoundation.org/

(ME-S) Osa Conservation https://osaconservation.org/

(ME-S) Sistema de Estudios de Posgrado of Universidad de Costa Rica https://www.sep.ucr.ac.cr/

4. We note that Figure 1 in your submission contain map image which may be copyrighted. All PLOS content is published under the Creative Commons Attribution License (CC BY 4.0), which means that the manuscript, images, and Supporting Information files will be freely available online, and any third party is permitted to access, download, copy, distribute, and use these materials in any way, even commercially, with proper attribution. For these reasons, we cannot publish previously copyrighted maps or satellite images created using proprietary data, such as Google software (Google Maps, Street View, and Earth). For more information, see our copyright guidelines: http://journals.plos.org/plosone/s/licenses-and-copyright.

Additional Editor Comments:

Thank you for submitting your manuscript PONE-D-22-04909 “Population structure and genetic connectivity of the scalloped hammerhead shark (Sphyrna lewini) across nursery grounds from the Eastern Tropical Pacific: implications for management and conservation” to PLOS ONE.

I have now received feedback from three experts in the field. As you can see below, they all found the paper of interest but had a few comments. In particular they found that some details on the analyses are missing which could make the paper more robust and understandable. They provided constructive comments that will help you to improve your paper.

As a result, I invite you to resubmit a revised version of the paper addressing the comments made by these referees.

With kind regards,

Johann

Reviewers' comments:

Reviewer's Responses to Questions

**Comments to the Author**

1. Is the manuscript technically sound, and do the data support the conclusions?

Reviewer #1: Yes

Reviewer #2: Yes

Reviewer #3: Partly

2. Has the statistical analysis been performed appropriately and rigorously? 

Reviewer #1: Yes

Reviewer #2: No

Reviewer #3: Yes

3. Have the authors made all data underlying the findings in their manuscript fully available?

Reviewer #1: Yes

Reviewer #2: No

Reviewer #3: Yes

4. Is the manuscript presented in an intelligible fashion and written in standard English?

Reviewer #1: Yes

Reviewer #2: Yes

Reviewer #3: Yes

5. Review Comments to the Author

Reviewer #1: PlosONE review Elizondo-Sancho et al. 2022

In this study, Elizondo-Sancho and collaborators add a significant number of samples to revisit the population structure of the Scalloped Hammerhead shark in the Eastern Tropical Pacific region. By doing so they provide valuable information for the management of this species. Overall, I found the manuscript well written and easy to follow. The genetic techniques used are not cutting-edge but the data is analysed adequately and I believe the conclusions are robust. I only made a few minor comments that I hope will improve the manuscript. Well done to all the authors. Pierre Feutry

L87-91: there are other philopatric shark species, look up genus Glyphis and Bull Shark

L112: are the acronyms ok for the locations in Costa Rica? There is a discrepancy between that sentence (COY, OJO and COS) and figure 1 (COY, OJO, ICO). Cocos Island seems to be missing in the locations from CR

L123-124: along the coast is quite vague and doesn’t really match up with the specificity of the sampling locations.

L149: “Nance et al. (2009 lewini” Typos?

L167 : add a « , » after « among haplotypes”

L191-192: related individuals should only be removed if there is a reason to think there was a bias in the sampling favoring the catch of related individuals, e.g. 2 neonates caught in the same eddy on the same day. On the opposite, if two siblings were caught 3 years apart there is no reason to remove one of them. See paper by Anderson & Waples (2017) for further details

L213-216: need to explain what method was used to choose the “best” K

L220-221: how did you deal with the potential of overfitting? Alpha score? Cross-validation?

L238: delete “relatively low”, this type of comment on the results belongs to the discussion.

L273: What is significant? Less than 0.05 p-value after correction? It’s generally better to provide the confidence interval for the Fst values rather than p-values, could add them in the upper diagonal

L284: delete “duly”

L305-306: move “(Fis)” after “inbreeding coefficients”

L309-311: replace “a greater” and “a lower” by “the greatest” and “the lowest” respectively

L312: replace “less” by “the least” and “greater” by “the most”

L368: most likely a bottleneck is not detected because the population size is not (yet) low enough, is that what you mean by “only a few generations have passed since overfishing started”. Please reword

L368-369: how can you report a “non-detectable bottleneck”? Do you mean that no bottleneck was detected?

L379: the entire mitogenome is one marker, I would reword and say “using more mitochondrial regions”

L380: Feutry et al. 2014 is a better reference for that statement

L404: or more markers, or both. See Foster et al. 2021

L415: not a challenge if you have plenty of markers (generally SNPs)

L430: quite likely there is further structuring, but more markers and/or more individuals are required to demonstrate it. I think the discussion need to emphasize a bit more the high potential for finer scale structure and the need to look into it with appropriate genetic techniques

References

Feutry, P., Kyne, P. M., Pillans, R. D., Chen, X., Naylor, G. J., & Grewe, P. M. (2014). Mitogenomics of the Speartooth Shark challenges ten years of control region sequencing. BMC evolutionary biology, 14(1), 1-9.

Foster, S. D., Feutry, P., Grewe, P., & Davies, C. (2021). Sample size requirements for genetic studies on yellowfin tuna. PloS one, 16(11), e0259113.

Waples, R. S. , & Anderson, E. C. (2017). Purging putative siblings from population genetic data sets: A cautionary view. Molecular Ecology, 26, 1211–1224. 10.1111/mec.14022

Reviewer #2: This research uses mtDNA and microsatellite data to assess the population structure of the Critically Endangered Scalloped Hammerhead Shark in the Eastern Tropical Pacific. These data build upon previous published datasets (which is wonderful!), revealing novel patterns of population structure. Given the highly threatened status of this species and the importance of population genetic data in conservation and management strategies, this is an important paper that need to be published. I am recommending major revisions on the basis that some of the information could be communicated more clearly through rephrasing, consistent use of terms, and some reorganization. I found some statements to be confusing as written, so I suggest rephrasing and being more concise to improve reader comprehension. The use of consistent names or acronyms for sampling sites would also improve clarity surrounding data analysis and results. Some of the information might flow better with some minor reorganization in the introduction and discussion. Specific suggestions regarding these points are below. The manuscript also needs a careful proofread. Comments/examples are provided in the manuscript pdf but these are not comprehensive.

The other reason I recommend major revision is that some data analyses are not presented, and this dataset may reveal additional novel and important information about the species with some further analysis. The data for HWE are not presented, so the p-values are unknown, as is the scale for these analyses. Was HWE significant calculated for all individuals pooled or for each sample site? This is important because of the assumptions of HWE (e.g. pooling two populations or not-pooling a single population could cause violations in the HWE assumptions). P values should also be reported for all statistics throughout. For example, Table 2 has ‘significant values in bold’, but the threshold used is not specified to give this more meaning. The authors state a correction was applied, so presumably it is not 0.05. Bottleneck tests are also mentioned, but no methods or results presented. Additional statistics that should be considered are: 1) exact tests, 2) DEST for mtDNA, 3) a hierarchical STRUCTURE analysis, 4) genetic diversity indices for the identified populations (rather than sampling sites), and 4) set up the AMOVA as a hypothesis. See specific comments below regarding some of these.

Line 64-65: All populations experience genetic drift, but the effects are more pronounced in small populations. Suggest rephrasing to make this clear.

Line 98: Suggest rephrasing from “sorting out the genetic diversity…”. For example, could change to “it is important to assess population structure and genetic diversity between potential nursery areas in this region”.

Line 100-103: Remove, as this is methods.

Line 112: Add sample sizes for each site.

Line 123: Here is says samples were collect from juveniles, but the introduction said YOY. Which is it? Similarly, what life stages were sampled in the other studies where the data is used here? This needs to be stated, and potential caveats addressed in the discussion since life stage sampled is needed to interpret the data more fully.

Line 153: The entire mtDNA control region?

Line 136: Remove “species-specific”. These primers may have been designed for S. lewini, but that does not mean they are species-specific. That would involve cross testing in other species to make sure they do not cross-amplify DNA from other species.

Lines 146, 159: More details are needed for how PCR products were cleaned and sequenced, as well as for fragment analysis, e.g. what size standard was used?

Line 156: Suggest including cycle numbers in Table S2 since they were not the same across all loci.

Line 165: Add sample sizes for these locations.

Line 173: Suggest exact tests here too.

Line 191-192: I understand why the authors removes FS from the analyses, but does this then impact population structure and genetic diversity statistics in the other direction? Since relatedness/sibship approaches can be used to elucidate population structure and natal philopatry, it might be worth exploring the data without this removal of FS and/or going further with relatedness analyses. See next comment as well.

Line 202-204: Were these comparisons also made between individuals at different sites? Suggest doing so to compare to the within-nursery data. It would also be interesting to look at sibship between different nursery areas as well as within nursery areas.

Line 210: Why not calculate DEST for mtDNA too?

Line 213: Suggest a hierarchical STRUCTURE if any of the identified populations (GUA, COS, PAN) have >1 sampling site. This is not really clear to me, as per the below comment for line 282. Also suggest looking at delta K

Table 1: Suggest calculating genetic diversity indices for each of the identified populations; this will give an estimate of diversity at scales relevant to management.

Line 250-252: Suggest rephrasing these sentences as they are confusing. I suspect the authors are trying to say that there were two common haplotypes across all sampling sites, but they were found at different frequencies in Mexico compared to Central America and Colombia.

Line 262-265: This statement is long and confusing. An AMOVA should be set up to test a specific hypothesis.

Line 282: I’m finding the acronyms and verbiage surrounding locations somewhat confusing. Here, GUA, COS, and PAN are mentioned. GUA and PAN are labelled on the map, but COS is not. I’m guessing that COS includes samples from >1 site, but the same may also be true for PAN based on the map. Later, there is reference to regions (e.g., 372) but it is difficult to follow given the inconsistencies. Suggest explaining sampling sites/ countries (pooled or not), regions (countries pooled?), etc. early on and then using the same language throughout the manuscript.

Line 285: “population-specific”- what was this defined by?

Line 291: Were all FS pairs from the same sampling site? This could be an interesting discussion point.

Line 308, 311, etc: Suggest reporting actual P-value. Was a correction applied to these statistical tests? If so, what was the new threshold?

Line 316-319: These statements seem to be contradictory. Were they all non-significant?

Line 321: Why were the samples from the Cocos Islands not included in analyses? For example, FST, DEST, Structure, etc.? The sample size wasn’t huge, but still worth including in the structure plot at a minimum.

Line 339-340: The statement “The average withing sampling site….” is confusing; suggest rephrasing.

Line 353 and elsewhere: Population declines can lead to a loss of genetic diversity, but that does not mean that: 1) population declines always cause declines in genetic diversity, or 2) that all populations with low diversity have undergone recent declines. This section seems to attribute the observed levels of genetic diversity to recent population declines, but this is not actually known. Elasmobranchs have some of the slowest rates of mutation among vertebrates, so genetic diversity accumulates slowly and can be low even in the absence of population declines. Suggest developing this section to be more comprehensive of genetic diversity in elasmobranchs, perhaps bringing in phylogeography (e.g. how recently might these populations been founded?)

Line 356: Levels of genetic diversity were not calculated for the central-southern ETP overall- they were calculated by sampling sites from what I can tell. Suggest analyzing genetic diversity for the identified populations to back up this statement. It also makes more sense from a management perspective to analyze data for each identified population.

Line 358-359: Take care with verbiage. Genetically distinct populations do not mean they resulted from independent evolutionary history. All populations of this species in the ETP likely have a common evolutionary history. This is evidenced by the presence of two common haplotypes shared across populations.

Line 367: It is stated that a bottleneck was not detected, but no data are presented to support this. Suggest either including bottleneck tests (with discussion on caveats of the various statistical approaches) or deleting the statement about the detection of bottleneck tests.

Line 373: The sentence on this line is confusing, suggest rephrasing.

Line 376: The term “sub-population’ has a specific meaning for the IUCN species assessments, which is cited as the source of this definition. The IUCN definition of ‘sub-population’ is not the same as used in population genetics. I suggest the authors read this definition more carefully and rework this point. The data presented in this paper does not support further splitting the EP sub-population of this species, as per the IUCN definition. Within this region, the identification of distinct population units is important to inform management, so suggest focusing on that.

Line 388-391; 397; 400-401; 436: The phrasing on these lines need some work as it is difficult to understand/follow. For example, line 388 mentions oceanographically dynamic regions and then uses the phrase ‘mixing zone’. Is this referring to a physical mixing zone or ‘mixing’ meaning gene flow? Line 400 mentions “high sampling effort” but not what locations fit this category. Etc.

Line 407: What age classes were sampled in these other studies?

Line 415: The discussion on relatedness could be built upon more. For example, what were the challenge for assessing sibship in this study? Could take some of the analysis further as well to support more discussion, as mentioned in previous comments.

Lines 423-429: This ought to be discussed in the population structure section. Philopatry is the logical explanation for the observed population structure, so integrate there. I also suggest either doing additional analyses on relatedness to develop this section more fully, or use the relatedness statistics to support the population structure findings.

Figure 1: Add sample sizes to caption or figure

Figure 2: It is difficult to see the ticks or count them.

Figure 5: What is the difference between the gray and black arrows? Specify in the caption.

Reviewer #3: In their article, Elizondo-Sancho describe the scalloped hammerhead’s population genetic structure and connectivity across nursery grounds from the Eastern Tropical Pacific, using a combination of mtDNA sequences and microsatellite markers. They employ commonly used population genetic analyses, and report stronger patterns of genetic structure than previously reported, suggestive of natal philopatry. They go on discussing the implications of these findings in terms of management of local populations.

Generally, the study is well conducted and I have no major issue with the analyses. Some important details (e.g. regarding STRUCTURE analyses) are missing, and I think some of the interpretations may be unwarranted. But these are fairly minor issues that I think can be very easily addressed by the authors with a minor revision. below I provide some detailed comments:

• Lines 50-51: I am not sure what the value of reporting Ho and allelic richness at microsatellite markers in the abstract is. These are very highly dependent on marker type (bialellic, tri-alellic, how where the markers selected).

• Line 87: i would rephrase as "allele frequency differences through time", since divergence usually refers to accumulation of mutations, while here the authors are talking of the effects of drift.

• Lines 214_216 and in general STRUCTURE analyses:

The authors do not explain how they chose the value for K they report in the results. What method was used to choose K (Evanno’s method? Other)? How do STRUCTURE plots look for different values of K? IS there any way to assess the admixture proportions? For SNPs data it's common to use evalAdmix (http://www.popgen.dk/software/index.php/EvalAdmix ), to evaluate pairwise correlation of residuals matrix between individuals. I am not sure whether there is an equivalent approach for microsatellite data.

Also, the admixture proportions reported in the figure are a bit difficult to reconcile with both the general population structure (Fst) and relative migration rates inferred: how is it that PAN and GUA show the lowest relative migration rates but the highest levels of admixture?

I am also not sure why the Authors have not reported structure analyses and DAPC of the entire dataset (including the samples from previous studies).

• DivMigrate analyses: what measure of genetic differentiation was used to estimate relative migration patterns ? Also please give more details on the method (including citation of the method implemented in divMigrate: Sunqvist et al 2016, Ecol Evol https://doi.org/10.1002/ece3.2096 ). How are relative migration rates scaled? i.e. is the highest migration rate given as 1?

An important concern is that this method assumes migration-drift equilibrium, I doubt this is a reasonable assumptions when it comes to long-lived marine animals with large Ne whose habitat has been affected by glaciations. See for example Maisano-Delser et al paper in Heredity on black-tip reef sharks and Walsh et al. paper in Heredity on grey reef sharks. So these results need to be interpreted with caution (as the authors of the package diveRsity themselves say).

• The authors mention low levels of genetic diversity (referring to pi and haplotype diversity). Low with respect to what? Other populations of the same species, other coastal sharks, or other marine fish? a pi of 0.0016 does not seem very low, but again this depends on what the reference is. What does seem interesting is the high degree of geographical heterogeneity in these estimates.

• Regarding migration rates, the authors mention that “Analysis of the extent and direction of gene flow showed no significant movement between coastal sampling sites.”. I am not sure how this conclusion was reached. There is no real analyses of the extent of gene flow, as measures of geneflow are "relative" (no absolute values ). Also the analyses assume migration-drift equilibrium and an island model, so the authors must be careful in interpreting the results.

• The authors mention that the low diversity of mtDNA is consistent with overexploitation. (Lines 352-353). No evidence is presented that the low levels of genetic diversity of this species are linked in any way to recent population declines. Given the generation time of scalloped hammerheads i find this hypothesis extremely unlikely.

None of the analyses the author presented allow any inference of recent changes in Ne, and to my knowledge such analyses would require extensive two-locus statistics (LD) obtained for a great portion of the genome, along with good linkage maps (for example, using the method developed by Santiago: https://doi.org/10.1093/molbev/msaa169) . Also please note that most studies on genetic diversity of sharks concluded that patterns of genetic diversity were almost certainly unrelated to recent population declines but rather reflect the species history of colonization/range expansion/isolation. See work on grey nurse sharks (Stow et al 2006 Biology Letters and subsequent paper in Molecular Ecology about grey nurse sharks https://doi.org/10.1098/rsbl.2006.0441
https://doi.org/10.1111/j.1365-294X.2009.04377.x , and recent work on blacktip reef shark by Stefano Mona and Maisano-Delser https://doi.org/10.1038/s41437-018-0164- , as well as work on grey reef sharks just published in heredity https://doi.org/10.1038/s41437-022-00514-4 ).

If the authors want to test this hypothesis they could try to use the R package “migraine” to detect possible bottlenecks, but they should also be aware that these estimates could be biased by complex demographic histories (e.g. https://doi.org/10.1038/s41437-018-0164-0 ).

• Lines 368-369: I am not sure what is meant by "non-detectable bottleneck effect". It could very well be that overharvesting may have reduced census size while having negligibly effects on effective population size. A non-detectable effect is not an effect at all?

6. PLOS authors have the option to publish the peer review history of their article (what does this mean?). If published, this will include your full peer review and any attached files.

Reviewer #1: **Yes: **Pierre Feutry

Reviewer #2: No

Reviewer #3: No

---

## [Author Response · Author response to Decision Letter 0]

23 Jun 2022

May 20th, 2022

San José, Costa Rica 

To Dr. Johann Mourier, Academic Editor;

Dr. Pierre Feutry, Referee; 

and two anonymous referees: 

First of all, thank you so much for your time to revise the manuscript. Your comments have been extremely helpful to improve this manuscript. Here below I address all of your comments. First I will start with observations about Journal Requirements and then I will address the comments regarding the content of the manuscript that were provided by each Referee. 

I look forward to you receiving the revised manuscript and expect that it now can meet PLOS ONE’s publication criteria. 

Kind regards, 

Mariana 

A. Regarding the comments about Journal Requirements: 

Yes thank you, I double checked that the manuscript has PLOS ONE’s style requirements.

2. We note that you have provided funding information. However, funding information should not appear in the Funding section or other areas of your manuscript.

Thank you, yes I eliminated the funding statement from the manuscript.

3. We note that you have stated that you will provide repository information for your data at acceptance. 

The data of mitochondrial control region sequences was deposited in Genbank and will not be released to the public database until December 2033,or until the data or accession numbers appear in print, whichever

is first.

4. We note that Figure 1 in your submission contain map image which may be copyrighted

I specified in the methods section that the map images were created by a public domain without any copyright. 

B. Regarding the comments about the content of the Manuscript: 

Reviewer #1: 

Thank you again Dr. Pierre Feutry, your comments were very welcome. Here I respond to your observations: 

1. L87-91: there are other philopatric shark species, look up genus Glyphis and Bull Shark

Thank you, I added these species to the introduction. 

2. L112: are the acronyms ok for the locations in Costa Rica? There is a discrepancy between that sentence (COY, OJO and COS) and figure 1 (COY, OJO, ICO). Cocos Island seems to be missing in the locations from CR

Yes, I missed mentioning Cocos Island in this section. For microsatellite analyses I only use the sampling site Ojochal of Costa Rica, that is why I was using the acronym OJO and COS for the same area, I understand this is confusing. I will change it. 

3. L123-124: along the coast is quite vague and doesn’t really match up with the specificity of the sampling locations.

Noted! And changed. 

4. L149: “Nance et al. (2009 lewini” Typos?

Yes! I did not see it before

5. L167 : add a « , » after « among haplotypes”

Yes, much better

6. L191-192: related individuals should only be removed if there is a reason to think there was a bias in the sampling favoring the catch of related individuals, e.g. 2 neonates caught in the same eddy on the same day. On the opposite, if two siblings were caught 3 years apart there is no reason to remove one of them. See paper by Anderson & Waples (2017) for further details

I ran all analyses with both data sets and the difference was very little, the total number of individuals that I removed from the analyses were 19, the majority were from Guatemala. No strong conclusion was different. Something that could further justify not removing these pairs of Full Siblings, is that most of them were sampled one year apart, and some even in different countries, so I would not trust that they are really full siblings. I checked the distribution of my data comparing it to what is expected of Unrelated pairs, Half Siblings, Full Sibling and Parent Offspring pairs (Fig 3). The majority are distributed as Unrelated, so I decided to leave all analyses with the full data set. 

7. L213-216: need to explain what method was used to choose the “best” K

Yes, the Evanno method was used

8. L220-221: how did you deal with the potential of overfitting? Alpha score? Cross-validation?

In the DAPC, retaining too many PCA axes with respect to the number of individuals can lead to over-fitting in the membership probabilities. To decide in an objective way how many PCA axes to retain, a cross validation analysis was performed with the xvalDapc function from the Adegenet package in R. This function tries different numbers of axes and the quality of the corresponding DAPC is assessed by cross-validation. The number of PCA axes associated with the lowest Mean Squared Error (40 PCs) were then retained in the DAPC. 

Jombart T, Devillard S and Balloux F (2010) Discriminant analysis of principal components: a new method for the analysis of genetically structured populations. BMC Genetics11:94. doi:10.1186/1471-2156-11-94

9. L238: delete “relatively low”, this type of comment on the results belongs to the discussion.

You are right, I will delete it

10. L273: What is significant? Less than 0.05 p-value after correction? It’s generally better to provide the confidence interval for the Fst values rather than p-values, could add them in the upper diagonal

The significance level was 0.05, so yes less tan 0.05 and there was no correction applied. For mitochondrial DNA Fst pairwise comparisons I could not find a way to provide the confidence intervals. I added the p-values to the upper diagonal. 

11. L284: delete “duly”

Deleted

12. L305-306: move “(Fis)” after “inbreeding coefficients”

Yes!

13. L309-311: replace “a greater” and “a lower” by “the greatest” and “the lowest” respectively

Thank you yes much better

14. L312: replace “less” by “the least” and “greater” by “the most”

Yes

15. L368: most likely a bottleneck is not detected because the population size is not (yet) low enough, is that what you mean by “only a few generations have passed since overfishing started”. Please reword

After consideration, I think that these results of the bottleneck analysis are not relevant. Other reviewers also suggested to eliminate this since it does not contribute much. 

16. L368-369: how can you report a “non-detectable bottleneck”? Do you mean that no bottleneck was detected?

After consideration, I think that these results of the bottleneck analysis are not relevant. Other reviewers also suggested to eliminate this since it does not contribute much. 

17. L379: the entire mitogenome is one marker, I would reword and say “using more mitochondrial regions”

Yes, this makes more sense

18. L380: Feutry et al. 2014 is a better reference for that statement

Yes!

19. L404: or more markers, or both. See Foster et al. 2021

Yes, I understand what you mean, it is important to use robust sample sizes or more markers or both. The thing is that here, I want to emphasize in that what out study has that is not seen in Nance et al., 2009 is a bigger sample size. 

20. L415: not a challenge if you have plenty of markers (generally SNPs)

Ok, I will specify genotypic data of few loci. 

21. L430: quite likely there is further structuring, but more markers and/or more individuals are required to demonstrate it. I think the discussion need to emphasize a bit more the high potential for finer scale structure and the need to look into it with appropriate genetic techniques 

Yes, I added this to the discussion

Reviewer #2

Thank you again for all your constructive comments, here I address each of them. And made the corresponding improvements to the manuscript. 

1. The data for HWE are not presented, so the p-values are unknown, as is the scale for these analyses. Was HWE significant calculated for all individuals pooled or for each sample site? This is important because of the assumptions of HWE (e.g. pooling two populations or not-pooling a single population could cause violations in the HWE assumptions). 

The data for HWE was calculated for all individuals pooled and for each sampling site separately, these results are presented in the Microsatellite loci section. There were no significant deviations from HWE after Holm-Bonferroni correction of the 9 loci that were used for analyses.

2. Line 64-65: All populations experience genetic drift, but the effects are more pronounced in small populations. Suggest rephrasing to make this clear.

You are right I rephrased it the following way: 

Population level declines are of major concern in conservation since the effects of genetic drift and inbreeding are pronounced in small populations, which may lead to loss of genetic diversity and compromise the ability of a population to cope with environmental change.

3. Line 98: Suggest rephrasing from “sorting out the genetic diversity…”. For example, could change to “it is important to assess population structure and genetic diversity between potential nursery areas in this region”.

Perfect, I rephrased this the following way: 

Given the limited data on the population structure of S. lewini and the high fishing pressure that this species is currently under throughout the ETP, it is important to assess sorting to assess population structure and genetic diversity in potential nursery areas of the region, to develop effective management and conservation strategies.

4. Line 100-103: Remove, as this is methods.

Yes, I removed those lines.

5. Line 112: Add sample sizes for each site.

Yes, I could do this but it would be a little confusing, since the samples that were used for the mitochondrial control region are not exactly the same as the ones used for the microsatellite loci analyses. I am going to add the total number of samples I had from each location

6. Line 123: Here is says samples were collect from juveniles, but the introduction said YOY. Which is it? Similarly, what life stages were sampled in the other studies where the data is used here? This needs to be stated, and potential caveats addressed in the discussion since life stage sampled is needed to interpret the data more fully.

The Castillo-Olguin and collaborators (2009) study sampled juveniles (60-130cm long).Nance and collaborators (2011) paper were juveniles 1-3 years old except adults in Manta Island (TL ≥ 1.5m). Quintanilla and collaborators (2015) juveniles 30-50 TL except Malpelo Island (adults TL ≥ 1.5m). Thank you I cleared this in methods

7. Line 153: The entire mtDNA control region?

Yes, this was modified

8. Line 136: Remove “species-specific”. These primers may have been designed for S. lewini, but that does not mean they are species-specific. That would involve cross testing in other species to make sure they do not cross-amplify DNA from other species.

Yes! You are right. Removed

9. Lines 146, 159: More details are needed for how PCR products were cleaned and sequenced, as well as for fragment analysis, e.g. what size standard was used?

Yes, for microsatellites I sequenced all loci in both directions to verify the microsatellite dinucleotide motifs. Then for fragment analyses it was done with a 5-dye chemistry (FAM, NED, PET, VIC) and an internal size standard LIZ GS500

10. Line 156: Suggest including cycle numbers in Table S2 since they were not the same across all loci.

Yes, this was from the initial protocol (8-20 cycles) I only used 8 cycles

11. Line 165: Add sample sizes for these locations.

Ready

12. Line 173: Suggest exact tests here too.

Yes, as you suggested I ran the exact test of population differentiation, in Arlequin. The results are almost exactly the same as the Fst values. I included the significance of the exact test in Table 2. 

13. Line 191-192: I understand why the authors removes FS from the analyses, but does this then impact population structure and genetic diversity statistics in the other direction? Since relatedness/sibship approaches can be used to elucidate population structure and natal philopatry, it might be worth exploring the data without this removal of FS and/or going further with relatedness analyses. See next comment as well.

Yes, I ran all analyses with both sets of data and the difference is minimal, no relevant conclusions changed. I explored a little further with relatedness analyses. 

Another reviewer made the observation that related individuals should only be removed if there is a reason to think there was a bias in the sampling favoring the catch of related individuals, e.g. 2 neonates caught in the same eddy on the same day. On the opposite, if two siblings were caught 3 years apart there is no reason to remove one of them. See paper by Anderson & Waples (2017) for further details. 

Something that could further justify not removing these pairs of Full Siblings, is that most of them were sampled one year apart, and some even in different countries, so I would not trust that they are really full siblings. 

14. Line 202-204: Were these comparisons also made between individuals at different sites? Suggest doing so to compare to the within-nursery data. It would also be interesting to look at sibship between different nursery areas as well as within nursery areas.

Yes, I added some analyses and graphs that explored further the relatedness within and between nursery areas. 

15. Line 210: Why not calculate DEST for mtDNA too?

I could not find any software or R package that would calculate Dest values for mtDNA data. 

16. Line 213: Suggest a hierarchical STRUCTURE if any of the identified populations (GUA, COS, PAN) have >1 sampling site. This is not really clear to me, as per the below comment for line 282. Also suggest looking at delta K

For this analyses there was only one sampling site per identified population. What I mean by this is only one sampling area in Guatemala, one in Costa Rica and one in Panamá. I also now present STRUCTURE plots with different values of K, and specify that the number of clusters identified by the Evanno method was K = 2. 

17. Table 1: Suggest calculating genetic diversity indices for each of the identified populations; this will give an estimate of diversity at scales relevant to management.

Yes very good idea, thank you I added this.

18. Line 250-252: Suggest rephrasing these sentences as they are confusing. I suspect the authors are trying to say that there were two common haplotypes across all sampling sites, but they were found at different frequencies in Mexico compared to Central America and Colombia.

Yes, I cleared this

19. Line 262-265: This statement is long and confusing. An AMOVA should be set up to test a specific hypothesis.

Yes, here I added different configurations of the data for the AMOVA analyses. The two group configuration (Northern ETP and Central-southern ETP), is the one that best explains the variation found. I specified the AMOVA as a hypothesis in the following way: 

In order to observe which configuration of the data best explained the variance, the AMOVA was performed with three different groupings: 1) one region (all locations); 2) two regions the Northern Eastern Tropical Pacific (Mexico) and the Central-southern Eastern Tropical Pacific (Guatemala, Costa Rica, Panama and Colombia); and 3) three regions the Northern Eastern Tropical Pacific (Mexico collection sites), the Central Eastern Pacific (Guatemala, Costa Rica and Panama), and the Southern Eastern Tropical Pacific (Colombia).

20. Line 282: I’m finding the acronyms and verbiage surrounding locations somewhat confusing. Here, GUA, COS, and PAN are mentioned. GUA and PAN are labelled on the map, but COS is not. I’m guessing that COS includes samples from >1 site, but the same may also be true for PAN based on the map. Later, there is reference to regions (e.g., 372) but it is difficult to follow given the inconsistencies. Suggest explaining sampling sites/ countries (pooled or not), regions (countries pooled?), etc. early on and then using the same language throughout the manuscript.

Yes, thank you I have cleared this misunderstanding. From now on I refer only to Ojochal (OJO), as it was the only location from Costa Rica that was analyzed with microsatellites.

21. Line 285: “population-specific”- what was this defined by?

Sorry yes, I meant sampling location. 

22. Line 291: Were all FS pairs from the same sampling site? This could be an interesting discussion point.

From the FS pairs that I found with 99% confidence, 17 were from the same sampling site. From the 5 that were from different sampling sites: 2 pairs were found between Costa Rica and Guatemala, and three pairs between Costa Rica and Panamá. I decided to not go further into these results, since there were inconsistencies when changing the alpha value. ML related identified as much as 200 pairs of FS in the data. This does not make sense, since they were pairs from different sampling sites and also from different years. I decided to maintain the analyses presented in Fig 3, in which the distribution of my data, is similar to that of Unrelated pairs (using the same allele frequencies as my data).

23. Line 308, 311, etc: Suggest reporting actual P-value. Was a correction applied to these statistical tests? If so, what was the new threshold?

Dest values and Weir and & Cockerham’s Fst values were calculated with the function fastDivPart in the R package diversity. The variance of these statistics was assessed by 100000 bootstrap iterations. For pairwise calculations carried out by the function, a bias corrected 95% confidence interval is calculated. I calculated the 95% confidence interval for these statistics 

24. Line 316-319: These statements seem to be contradictory. Were they all non-significant?

Yes, this was written in a confusing way. What this analysis did was test if there was asymmetric gene flow between these areas, which was not found. The statement should be corrected the following way: 

Analysis of the extent and direction of gene flow showed no significant asymmetric movement between coastal sampling sites.

25. Line 321: Why were the samples from the Cocos Islands not included in analyses? For example, FST, DEST, Structure, etc.? The sample size wasn’t huge, but still worth including in the structure plot at a minimum.

Yes, I will ran these analyses with samples from Cocos included, and now present these results.

26. Line 339-340: The statement “The average withing sampling site….” is confusing; suggest rephrasing.

Thank you, yes. This should be: The average relatedness examined within sampling sites and overall sampling sites is the statistic test (observed in a red arrow). 

27. Line 353 and elsewhere: Population declines can lead to a loss of genetic diversity, but that does not mean that: 1) population declines always cause declines in genetic diversity, or 2) that all populations with low diversity have undergone recent declines. This section seems to attribute the observed levels of genetic diversity to recent population declines, but this is not actually known. Elasmobranchs have some of the slowest rates of mutation among vertebrates, so genetic diversity accumulates slowly and can be low even in the absence of population declines. Suggest developing this section to be more comprehensive of genetic diversity in elasmobranchs, perhaps bringing in phylogeography (e.g. how recently might these populations been founded?)

Yes, I re-wrote this part of the Discussion.

28. Line 356: Levels of genetic diversity were not calculated for the central-southern ETP overall- they were calculated by sampling sites from what I can tell. Suggest analyzing genetic diversity for the identified populations to back up this statement. It also makes more sense from a management perspective to analyze data for each identified population.

Yes! I added these results.

29. Line 358-359: Take care with verbiage. Genetically distinct populations do not mean they resulted from independent evolutionary history. All populations of this species in the ETP likely have a common evolutionary history. This is evidenced by the presence of two common haplotypes shared across populations.

Yes, I re-wrote this part of the Discussion. 

30. Line 367: It is stated that a bottleneck was not detected, but no data are presented to support this. Suggest either including bottleneck tests (with discussion on caveats of the various statistical approaches) or deleting the statement about the detection of bottleneck tests.

Yes, I consider deleting the statement is better. This is not a relevant result.

31. Line 373: The sentence on this line is confusing, suggest rephrasing.

Yes, I rephrased it the following way: 

This pattern is mainly due to an uneven distribution of the two most common haplotypes, one is found in higher frequency in the Northern ETP while the other is found in higher frequency in the Central-southern ETP.

32. Line 376: The term “sub-population’ has a specific meaning for the IUCN species assessments, which is cited as the source of this definition. The IUCN definition of ‘sub-population’ is not the same as used in population genetics. I suggest the authors read this definition more carefully and rework this point. The data presented in this paper does not support further splitting the EP sub-population of this species, as per the IUCN definition. Within this region, the identification of distinct population units is important to inform management, so suggest focusing on that.

Thank you, yes I had not considered the strict definition of subpopulation of the IUCN, which specifies that they are groups with very little exchange. In this case I was referring specifically to a distinct population segments, defined under the Endangered Species Act as a vertebrate population or group of populations that is discrete from other populations of the species and significant in relation to the entire species. 

33. Line 388-391; 397; 400-401; 436: The phrasing on these lines need some work as it is difficult to understand/follow. For example, line 388 mentions oceanographically dynamic regions and then uses the phrase ‘mixing zone’. Is this referring to a physical mixing zone or ‘mixing’ meaning gene flow? 

Yes, in this study (Rodríguez-Zárate et al., 2018) make a simulation of what would happen if they freed a particle from a coastal area along the coast of Central America and Mexico, where would the particle drift depending on oceanographic conditions, so here they are referring to a physical mixing zone. 

34. Line 400 mentions “high sampling effort” but not what locations fit this category. Etc.

Yes I rephrased this. 

35. Line 407: What age classes were sampled in these other studies?

Thank you yes, I added this to the methods section. The Castillo-Olguin and colaborators (2009) study sampled juveniles (60-130cm long).Nance et al. (2011) paper were juveniles 1-3 years old except adults in Manta Island (TL ≥ 1.5m). Quintanilla et al. (2015) juveniles 30-50 TL except Malpelo Island (adults TL ≥ 1.5m). 

36. Line 415: The discussion on relatedness could be built upon more. For example, what were the challenge for assessing sibship in this study? Could take some of the analysis further as well to support more discussion, as mentioned in previous comments.

Yes, I took some of these analyses further. 

37. Lines 423-429: This ought to be discussed in the population structure section. Philopatry is the logical explanation for the observed population structure, so integrate there. I also suggest either doing additional analyses on relatedness to develop this section more fully, or use the relatedness statistics to support the population structure findings.

I understand that this paragraph relates to the population structure section. Philopatry in this case is generating the population structure observed, so I still consider that is should be in the section of Relatedness and Natal Philopatry. 

38. Figure 1: Add sample sizes to caption or figure 

Added

39. Figure 2: It is difficult to see the ticks or count them. 

Yes, I added some lines for them to be more visible

40. Figure 5: What is the difference between the gray and black arrows? Specify in the caption. 

Ready, the darker the arrows the higher the relative gene flow. 

Reviewer #3. 

Thank you so much for your valuable comments, here I address each of them: 

1. Lines 50-51: I am not sure what the value of reporting Ho and allelic richness at microsatellite markers in the abstract is. These are very highly dependent on marker type (bialellic, tri-alellic, how where the markers selected).

You are right, this is not a very substantial result, I will eliminate it from the abstract. 

2. Line 87: i would rephrase as "allele frequency differences through time", since divergence usually refers to accumulation of mutations, while here the authors are talking of the effects of drift.

I agree. I will rephrase it with the word differentiation instead of divergence. 

3. Lines 214_216 and in general STRUCTURE analyses:The authors do not explain how they chose the value for K they report in the results. What method was used to choose K (Evanno’s method? Other)? 

Yes, Evanno’s method was used to choose K 

4. How do STRUCTURE plots look for different values of K? IS there any way to assess the admixture proportions? For SNPs data it's common to use evalAdmix (http://www.popgen.dk/software/index.php/EvalAdmix ), to evaluate pairwise correlation of residuals matrix between individuals. I am not sure whether there is an equivalent approach for microsatellite data.

Also, the admixture proportions reported in the figure are a bit difficult to reconcile with both the general population structure (Fst) and relative migration rates inferred: how is it that PAN and GUA show the lowest relative migration rates but the highest levels of admixture?

I re-ran the STRUCTURE analyses and present the plots for different values of K as you suggested. Levels of admixture now coincide with other population structure analyses. 

5. I am also not sure why the Authors have not reported structure analyses and DAPC of the entire dataset (including the samples from previous studies).

Structure and DAPC analyses were done only for the locations where I had microsatellite loci genotype data. This was only for the locations of Costa Rica: Ojochal (OJO) and Cocos Island (ICO); Guatemala (GUA) and Panamá (PAN)

6. DivMigrate analyses: what measure of genetic differentiation was used to estimate relative migration patterns?

Dest was used to estimate relative migration patterns

7. Also please give more details on the method (including citation of the method implemented in divMigrate: Sunqvist et al 2016, Ecol Evol https://doi.org/10.1002/ece3.2096 ). How are relative migration rates scaled? i.e. is the highest migration rate given as 1? An important concern is that this method assumes migration-drift equilibrium, I doubt this is a reasonable assumptions when it comes to long-lived marine animals with large Ne whose habitat has been affected by glaciations. See for example Maisano-Delser et al paper in Heredity on black-tip reef sharks and Walsh et al. paper in Heredity on grey reef sharks. So these results need to be interpreted with caution (as the authors of the package diveRsity themselves say).

Yes, thank you I provide more details on this method. The “divMigrate” function was used to plot the relative migration levels and detect asymmetries in gene flow patterns, between pairs of population samples using DEST values of genetic differentiation (Sundqvist et al., 2016). This function plots sampling areas connected to every other by two connections that represent the two reciprocal gene flow components between any pair of locations (Sundqvist et al., 2016). This approach provides information on the direction of migration using relative migration scales (from 0 to 1) in which the highest migration rate given is 1 (Sundqvist et al., 2016).

Yes, these articles you mention make a thorough analysis of past and present demographics of these species. They have more statistical power, since they are using a large genomic data set of SNPs. Even with this statistical power, these authors mention how equilibrium models of population structure are not realistic and can give misleading results. They mention how increases in Ne can be confounded by increases in migration. They mention how population bottlenecks can also be mistaken for what really is a low number of migrants associated to a metapopulation structure. The genetic variability produces a similar pattern. Thank you for mentioning them to me, but I consider this analyses are a way to visualize other population structure analyses like DAPC, genetic differentiation indexes and STRUCTURE analyses. This graphs provide an idea of how much variation is shared between locations. Also knowing that sampling sizes are different, for example between Cocos Island (N = 15) and areas of the coast (N = 50) there is a big difference of sample sizes, this could be affecting the way connectivity is observed, being highest from the island to the coast, and lowest from the coast to the island. Still I mention in the results that the asymmetry of gene flow is not significant. 

8. The authors mention low levels of genetic diversity (referring to pi and haplotype diversity). Low with respect to what? Other populations of the same species, other coastal sharks, or other marine fish? a pi of 0.0016 does not seem very low, but again this depends on what the reference is. What does seem interesting is the high degree of geographical heterogeneity in these estimates.

I re-wrote this section of the discussion. These values can be compared to other related species of sharks from the ETP, like Spyrna zygaena that has Haplotype diversity of 0.86 and nucleotide diversity of 0.26 (Feliz-Lopez., 2019). This species in the south pacific h=0.615 (Hernández., 2013). Sphyrna tiburo h = 0.932 Escatel-Luna., 2015. And S. lewini in the ETP with previous studies h=0.53 Nance et al., 2011 and Castillo-Olguin h=0.49. 

9. Regarding migration rates, the authors mention that “Analysis of the extent and direction of gene flow showed no significant movement between coastal sampling sites.”. I am not sure how this conclusion was reached. There is no real analyses of the extent of gene flow, as measures of geneflow are "relative" (no absolute values ). Also the analyses assume migration-drift equilibrium and an island model, so the authors must be careful in interpreting the results.

Yes, this was written in a confusing way. What this analysis did was test if there was asymmetric gene flow between these areas, which was not found. The statement should be corrected the following way:

Analysis of the extent and direction of gene flow showed no significant asymmetric movement between coastal sampling sites.

10. The authors mention that the low diversity of mtDNA is consistent with overexploitation. (Lines 352-353). No evidence is presented that the low levels of genetic diversity of this species are linked in any way to recent population declines. Given the generation time of scalloped hammerheads i find this hypothesis extremely unlikely.

None of the analyses the author presented allow any inference of recent changes in Ne, and to my knowledge such analyses would require extensive two-locus statistics (LD) obtained for a great portion of the genome, along with good linkage maps (for example, using the method developed by Santiago: https://doi.org/10.1093/molbev/msaa169) . Also please note that most studies on genetic diversity of sharks concluded that patterns of genetic diversity were almost certainly unrelated to recent population declines but rather reflect the species history of colonization/range expansion/isolation. See work on grey nurse sharks (Stow et al 2006 Biology Letters and subsequent paper in Molecular Ecology about grey nurse sharks https://doi.org/10.1098/rsbl.2006.0441
https://doi.org/10.1111/j.1365-294X.2009.04377.x , and recent work on blacktip reef shark by Stefano Mona and Maisano-Delser https://doi.org/10.1038/s41437-018-0164- , as well as work on grey reef sharks just published in heredity https://doi.org/10.1038/s41437-022-00514-4 ).

If the authors want to test this hypothesis they could try to use the R package “migraine” to detect possible bottlenecks, but they should also be aware that these estimates could be biased by complex demographic histories (e.g. https://doi.org/10.1038/s41437-018-0164-0 ). 

Thank you, I addressed these observations in the first paragraph of the Genetic diversity part of the Discussion 

11. Lines 368-369: I am not sure what is meant by "non-detectable bottleneck effect". It could very well be that overharvesting may have reduced census size while having negligibly effects on effective population size. A non-detectable effect is not an effect at all?

Yes, I realize this is not clear. When I did the bottleneck analyses, there was none detected. So yes, it is not an effect. After consideration, I think that these results of the bottleneck analysis are not relevant. Other reviewers also suggested to eliminate this since it does not contribute much.

---

## [Decision Letter · Decision Letter 1]

26 Jul 2022

PONE-D-22-04909R1Population structure and genetic connectivity of the scalloped hammerhead shark (Sphyrna lewini) across nursery grounds from the Eastern Tropical Pacific: implications for management and conservationPLOS ONE

Dear Dr. Elizondo-Sancho,

Thank you for submitting your manuscript to PLOS ONE. After careful consideration, we feel that it has merit but does not fully meet PLOS ONE’s publication criteria as it currently stands. Therefore, we invite you to submit a revised version of the manuscript that addresses the points raised during the review process.

Dear Mariana Elizondo-Sancho,

I have now received the comments of one of the original reviewer who only highlighted a minor point to address.

Please address it in a revised version and I will be happy to accept your work for publication in Plos One.

Kind regards

Johann

We look forward to receiving your revised manuscript.

Kind regards,

Johann Mourier, Ph.D.

Academic Editor

PLOS ONE

Journal Requirements:

Additional Editor Comments:

Dear Mariana Elizondo-Sancho,

I have now received the comments of one of the original reviewer who only highlighted a minor point to address.

Please address it in a revised version and I will be happy to accept your work for publication in Plos One.

Kind regards

Johann

Reviewers' comments:

Reviewer's Responses to Questions

**Comments to the Author**

1. If the authors have adequately addressed your comments raised in a previous round of review and you feel that this manuscript is now acceptable for publication, you may indicate that here to bypass the “Comments to the Author” section, enter your conflict of interest statement in the “Confidential to Editor” section, and submit your "Accept" recommendation.

Reviewer #3: All comments have been addressed

2. Is the manuscript technically sound, and do the data support the conclusions?

Reviewer #3: Yes

3. Has the statistical analysis been performed appropriately and rigorously? 

Reviewer #3: Yes

4. Have the authors made all data underlying the findings in their manuscript fully available?

Reviewer #3: Yes

5. Is the manuscript presented in an intelligible fashion and written in standard English?

Reviewer #3: Yes

6. Review Comments to the Author

Reviewer #3: The authors addressed most of my concerns.

A further comment to be addressed in a minor revision (no need to send this back to me for review)

In the discussion about genetic diversity, the authors mention the slow mutation rate for mtDNA in sharks. Note that genetic diversity is a product of Ne and mutation rate, i.e. nucleotide diversity is determined by the mutation scaled population size (or the population scaled mutation rate, if you prefer). So that at equilibrium pi= = 4Ne (where is the mutation rate). So the authors are correct that the mutation rate alone does not explain low nucleotide diversity, but this does not suggest at all that nucleotide diversity could reflect overexploitation. It most likely (given generation time and time for pi to reach equilibrium) reflects historical demographic events. So the point that diversity is likely shaped by long term Ne or other demographic events should be made, in my opinion. Indeed, theta (and pi at equilibrium) are indeed a measure of population size given a mutation rate.

7. PLOS authors have the option to publish the peer review history of their article (what does this mean?). If published, this will include your full peer review and any attached files.

Reviewer #3: No

---

## [Author Response · Author response to Decision Letter 1]

17 Aug 2022

Reviewer #3: The authors addressed most of my concerns.

A further comment to be addressed in a minor revision (no need to send this back to me for review)

In the discussion about genetic diversity, the authors mention the slow mutation rate for mtDNA in sharks. Note that genetic diversity is a product of Ne and mutation rate, i.e. nucleotide diversity is determined by the mutation scaled population size (or the population scaled mutation rate, if you prefer). So that at equilibrium pi= = 4Ne (where is the mutation rate). So the authors are correct that the mutation rate alone does not explain low nucleotide diversity, but this does not suggest at all that nucleotide diversity could reflect overexploitation. It most likely (given generation time and time for pi to reach equilibrium) reflects historical demographic events. So the point that diversity is likely shaped by long term Ne or other demographic events should be made, in my opinion. Indeed, theta (and pi at equilibrium) are indeed a measure of population size given a mutation rate

Thank you, I addressed all the suggestions of low nucleotide diversity reflecting overexploitation in the Manuscript. In order to do so, I eliminated the following citations from the reference list:

78. Chapman DD, Pinhal D, Shivji MS. Tracking the fin trade: Genetic stock identification in western Atlantic scalloped hammerhead sharks Sphyrna lewini. Endanger Species Res. 2009; 

79. Baum JK, Myers RA, Kehler DG, Worm B, Harley SJ, Doherty PA. Collapse and conservation of shark populations in the Northwest Atlantic. Science (80- ). 2003;

---

## [Editor Report · Decision Letter 2]

19 Aug 2022

Population structure and genetic connectivity of the scalloped hammerhead shark (Sphyrna lewini) across nursery grounds from the Eastern Tropical Pacific: implications for management and conservation

PONE-D-22-04909R2

Dear Dr. Elizondo-Sancho,

We’re pleased to inform you that your manuscript has been judged scientifically suitable for publication and will be formally accepted for publication once it meets all outstanding technical requirements.

Kind regards,

Johann Mourier, Ph.D.

Academic Editor

PLOS ONE

Additional Editor Comments (optional):

Thank you for your effort in revising your manuscript.

I am now happy to recommend your work to be published in Plos One. I hope you enjoyed the reviewing process to help you improve your manuscript.

Kind regards

Johann
---

## [Editor Report · Acceptance letter]

2 Sep 2022

PONE-D-22-04909R2 

Population structure and genetic connectivity of the scalloped hammerhead shark (*Sphyrna lewini*) across nursery grounds from the Eastern Tropical Pacific: implications for management and conservation 

Dear Dr. Elizondo-Sancho:

I'm pleased to inform you that your manuscript has been deemed suitable for publication in PLOS ONE. Congratulations! Your manuscript is now with our production department. 

Kind regards, 

on behalf of

Dr. Johann Mourier 

Academic Editor

PLOS ONE